# The diagnostic potential of proteomics and machine learning in Lyme neuroborreliosis

Annelaura Bach Nielsen [1,2,10], Lasse Fjordside[3,10], Lylia Drici [2], Maud Eline Ottenheijm[1,2], Christine Rasmussen[1], Anna J. Henningsson [4,5], Lene Holm Harritshøj[6,7], Matthias Mann [2,8], Helene Mens[3], Anne-Mette Lebech [3,7,11] & Nicolai J. Wewer Albrechtsen [1,7,9,11] ✉

Lyme neuroborreliosis (LNB), a nervous system infection caused by tick-borne spirochetes of the *Borrelia burgdorferi* sensu lato complex, is among the most frequent bacterial infections of the nervous system in Europe. Early diagnosis and continuous monitoring remain challenging due to limited sensitivity and specificity of current methods and requires invasive lumbar punctures, underscoring the need for improved, less invasive diagnostic tools. Here, we apply mass spectrometry-based proteomics to analyse 308 cerebrospinal fluid (CSF) samples and 207 plasma samples from patients with LNB, viral meningitis, controls and other manifestations of Lyme borreliosis. Diagnostic panels of regulated proteins are identified and evaluated through machine learning-assisted proteome analyses. In CSF, the classifier distinguishes LNB from viral meningitis and controls with AUCs of 0.92 and 0.90, respectively. In plasma, LNB is distinguished from controls with an AUC of 0.80. Our findings suggest a potential diagnostic role for machine learning-assisted proteomics in adults with LNB.

Lyme neuroborreliosis (LNB) is a bacterial infection of the nervous system caused by spirochetes of the *Borrelia burgdorferi* sensu lato (*B. burgdorferi* s.l.) complex transmitted through bites from hard-shelled ticks of the *Ixodes* genus[1]. With an approximate incidence in endemic countries between 3.2 and 6.3 per 100,000 persons/year[2–4] LNB is among the most frequent bacterial infections of the nervous system in Europe[5,6]. LNB can cause a wide range of clinical neurological conditions, but most frequently presents as a subacute painful meningoradiculitis with radiating pain from the spine to neck, extremities, thorax or abdomen, lymphocytic meningitis and/or cranial

neuropathies, e.g., facial nerve palsy[7]. If relevant antibiotic therapy is administered at an early stage of disease, LNB has a favorable long-term prognosis[8–10]. However, delayed treatment is associated with an increased risk of residual symptoms and long-term sequelae[9]. In countries where LNB is endemic, the average time from onset of neurological symptoms to diagnosis is typically around 3 weeks and has remained unchanged for the last four decades[6].

The cause of the diagnostic delay is multifactorial. Overlapping symptomatology with other more common diseases and the fact that only around 40% of patients with LNB report a tick bite and only 25%

[1]Department of Clinical Biochemistry, Copenhagen University Hospital - Bispebjerg and Frederiksberg Hospital, Copenhagen, Denmark. [2]NNF Center for Protein Research, Faculty of Health and Medical Sciences, University of Copenhagen, Copenhagen, Denmark. [3]Department of Infectious Diseases, Copenhagen University Hospital, Rigshospitalet, Copenhagen, Denmark. [4]Department of Biomedical and Clinical Sciences, Linköping University, Linköping, Sweden. [5]National Reference Laboratory for Tick-Borne Bacteria, Department of Laboratory Medicine, Region Jönköping County, Jönköping, Sweden. [6]Department of Clinical Immunology, Copenhagen University Hospital, Copenhagen, Denmark. [7]Department of Clinical Medicine, Faculty of Health and Medical Sciences, University of Copenhagen, Copenhagen, Denmark. [8]Department of Proteomics and Signal Transduction, Max Planck Institute of Biochemistry, Martinsried, Germany. [9]Copenhagen Center for Translational Research, Copenhagen University Hospital - Bispebjerg and Frederiksberg Hospital, Copenhagen, Denmark. [10]These authors contributed equally: Annelaura Bach Nielsen, Lasse Fjordside. [11]These authors jointly supervised this work: Anne-Mette Lebech, Nicolai J. Wewer Albrechtsen. ✉e-mail: nicolai.albrechtsen@regionh.dk

report a history of the classic skin rash erythema migrans make it less likely that physicians and patients consider LNB as a differential diagnosis, especially in the absence of facial nerve palsy[6,8].

However, even when LNB is clinically suspected at an early stage, analyses of cerebrospinal fluid (CSF) are currently required to establish the diagnosis. A lumbar puncture is an uncomfortable and expensive procedure that may require hospitalization and general anesthesia for children. Direct pathogen identification of *B. burgdorferi* s.l. with polymerase chain reaction (PCR), cultivation, or a combination has retained an exceedingly poor level of sensitivity both in blood and CSF despite several attempts to improve tools and techniques over the years[11–14]. Further, the serological response in LNB is not detectable early in the course of disease, and antibodies may remain elevated for months to several years after full recovery of well-treated infections, making it impossible to discriminate between past and current infection. Furthermore, the *B. burgdorferi* s.l.-specific IgG and IgM in blood poorly predict nervous system involvement[15].

Thus, in order to reduce the diagnostic delay in LNB, the need for novel diagnostic tools is evident. Especially improved blood-based diagnostic tests would be extremely valuable as they would both potentially help reduce the diagnostic delay and offer a less invasive diagnostic tool.

Untargeted proteomic analysis enables accurate measurement of proteins in a sample and can be used to compare patterns of protein regulation in patients with and without disease to identify disease-specific protein response signatures[16]. When combined with machine learning for data evaluation and analysis, proteomic test results can be delivered within hours, making it highly attractive in real-life clinical settings.

The aim of this study was to characterize host-response protein signatures in adult patients with LNB using mass-spectrometry-based proteomics, and to explore whether machine-learning-assisted analysis of these signatures could contribute to the development of novel diagnostic tools for LNB in CSF and plasma.

## Table 1 | Baseline characteristics of patients with Lyme neuroborreliosis, viral meningitis, controls, and other Lyme borreliosis manifestations

| Cohort | Diagnosis | n | Age mean [IQR] | Females n (%) |
|---|---|---|---|---|
| CSF development | Lyme neuroborreliosis | 49 | 51 [40–66] | 26 (53) |
| N = 145 | Viral meningitis | 44 | 38 [27–42] | 21 (47) |
|  | Control | 52 | 36 [25–42] | 17 (32) |
| CSF validation | Lyme neuroborreliosis | 69 | 53 [46–67] | 28 (40) |
| N = 163 | Viral meningitis | 45 | 31 [25–37] | 23 (51) |
|  | Control | 49 | 44 [33–53] | 24 (48) |
| Plasma development | Lyme neuroborreliosis | 27 | 64 [61–69] | 10 (37) |
| N = 95 | Viral meningitis | 20 | 35 [27–38] | 13 (65) |
|  | Control | 48 | 46 [33–60] | 28 (58) |
| Plasma validation | Lyme neuroborreliosis | 10 | 59 [56–63] | 4 (40) |
| N = 80 | Viral meningitis | 18 | 38 [24–43] | 10 (55) |
|  | Control | 20 | 36 [28–42] | 6 (30) |
|  | Acrodermatitis chronica atrophicans | 6 | 66 [59–74] | 6 (100) |
|  | Erythema migrans | 9 | 47 [36–54] | 5 (55) |
|  | Post-treatment-Lyme-disease-syndrome | 17 | 58 [48–64] | 10 (58) |

*CSF* cerebrospinal fluid, *IQR* inter quartile range.

## Results

### Study population
A total of 483 CSF and plasma samples from adult individuals were eligible for proteomics analysis: 155 samples from patients with LNB, 127 samples from patients with viral meningitis, 169 controls, and 32 samples from patients with erythema migrans, acrodermatitis chronica atrophicans, and post-treatment-Lyme-disease-syndrome (Table 1).

### CSF proteomics
**CSF development cohort.** A total of 1865 proteins were identified in CSF, with a median of 773 quantified per sample (Supplementary Fig. 1). Fourteen samples were excluded due to low protein numbers. After filtering for missingness, a total of 654 proteins were included for further analysis.

A total of 176 proteins were significantly different between LNB and viral meningitis; of these, 10 had an absolute log2 fold change larger than 1 (Fig. 1A, Supplementary Table 1). A total of 464 proteins were significantly different between LNB and controls; of these, 41 had an absolute log2 fold change larger than 1 (Fig. 1A, Supplementary Table 1).

Of the significant proteins with high log2 fold changes in the two comparisons, 7 proteins overlapped (Fig. 1B).

**CSF validation cohort.** In the validation cohort, a total of 147 proteins were significantly different in LNB compared to viral meningitis and controls with high log2 fold changes, where 41/44 proteins from the development cohort overlapped (Fig. 1B and Supplementary Figs. 2 and 3).

**Protein signatures of LNB in CSF.** Protein signatures were dominated by immunoglobulins (e.g., IGLV3-25, IGLV2–18, FCGBP, IGHM), proteins involved in innate immune responses (e.g., enolase 1, S100A9), neuroendocrine signaling (e.g., ECRG4, CHGA), cell migration and cell damage (e.g., PFN1, APCS, YWHAZ, H4C1, ACTA2) (Fig. 1C). Immunoglobulins were generally upregulated, while proteins involved in innate immune responses and neuroendocrine signaling were relatively downregulated in LNB compared to viral meningitis and controls.

### Diagnostic classifier for LNB based on machine learning and CSF proteomics
The protein profiles of the CSF development cohort were used to develop a machine-learning model to evaluate the potential of mass-spectrometry-based proteomics coupled with machine learning as diagnostic support in LNB. Of the twelve machine-learning models tested, the Logistic Regression (LR) model obtained the highest performance in the classification of viral meningitis vs. LNB, whereas the Support Vector Classifier (SVC) model produced the best result in the control vs. LNB classification (Supplementary Figs. 4 and 5, respectively). The performance of the diagnostic classification model of viral meningitis vs. LNB on the test set had an area under the curve (AUC) of 0.91 (std. = 0.11) and a Matthews Correlation Coefficient (MCC) of 0.81 (std. = 0.08) (Fig. 2A). When applied on the validation cohort, the model obtained an AUC and MCC of 0.92 (std. = 0.02) and 0.7 (std. = 0.06). According to their SHapley Additive exPlanations (SHAP) values, the most important proteins in the discrimination between viral meningitis and LNB are visualized in Fig. 2B. The performance of the diagnostic classification model of control vs. LNB on the test set had an AUC of 0.93 (std. = 0.07) and a MCC of 0.76 (std. = 0.14) (Fig. 2C). When applied on the validation cohort, the model obtained an AUC and MCC of 0.9 (std. = 0.01) and 0.63 (std. = 0.06). The most important proteins in the model, according to their SHAP values

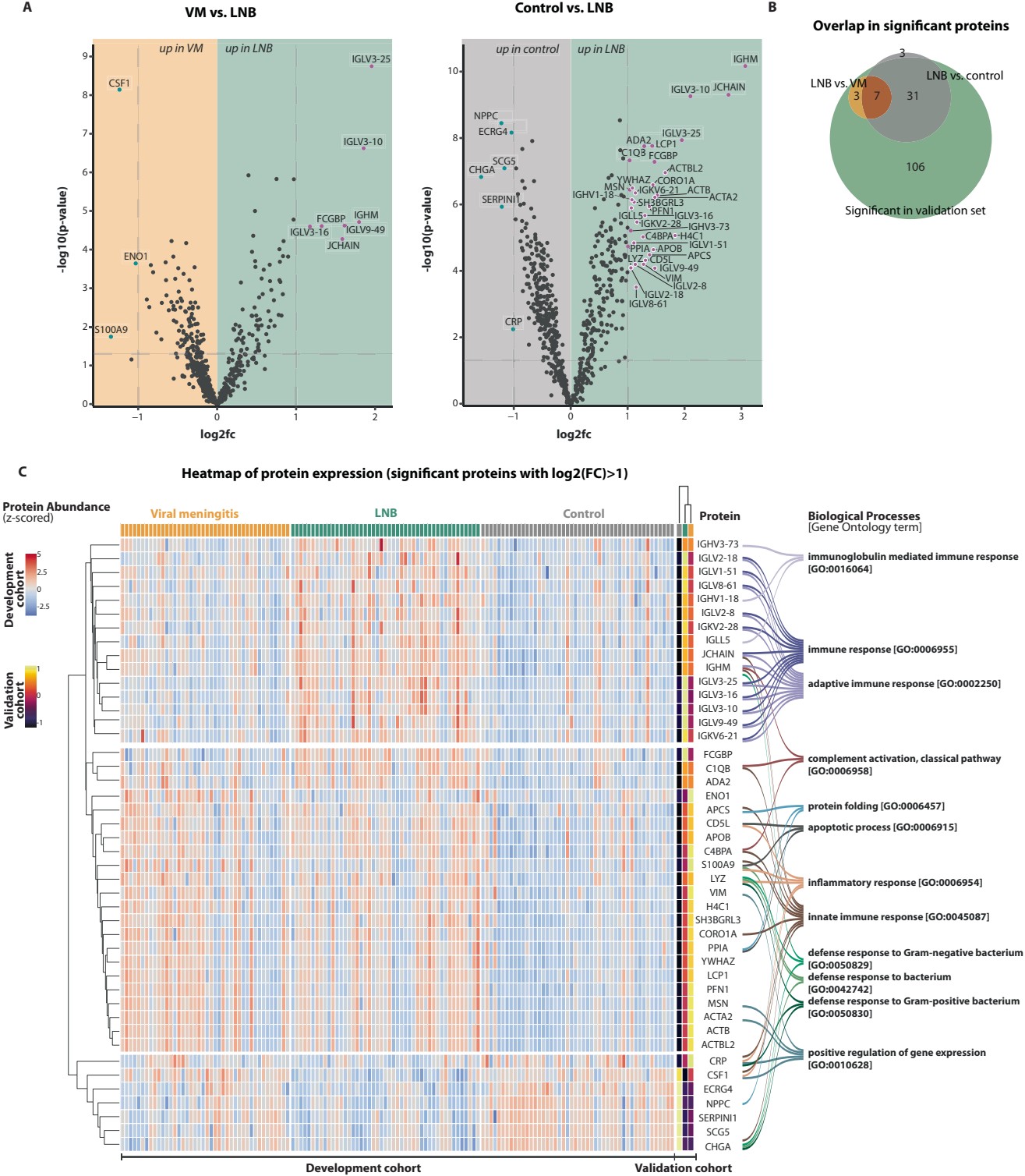

**Fig. 1 | Proteome changes in the cerebrospinal fluid (CSF) of Lyme Neuroborreliosis (LNB) patients. A** Volcano plots highlighting protein differences between Lyme neuroborreliosis (LNB) vs. viral meningitis (VM) and LNB vs. Controls in the cerebrospinal fluid development cohort. Significant proteins were identified by a two-tailed *t*-test adjusted for multiple hypothesis testing with Benjamini–Hochberg correction; adjusted *p* values < 0.05 were deemed statistically significant. Each point represents a protein; the color purple represents upregulated proteins with a log2 fold change (FC) larger than 1, the color blue represents downregulated proteins with a log2 fold smaller than −1, and the color black represents proteins with a log2 fold change between −1 and 1 or non-significant proteins. *X*-axis depicts the log2 fold change, and *y*-axis depicts the log10 adjusted *p* value. Horizontal dashed lines represent the significance threshold after multiple hypothesis correction at *p* = 0.05. Vertical dashed lines represent large log2 fold changes: above 1 or below −1. **B** Overlap in significant proteins from the two volcano plots with log2 fold changes lower than −1 or larger than 1. **C** Heatmap of the significant proteins for each sample in the development cohort and the corresponding mean from the validation cohort. The Gene Ontology term associated with the protein is depicted on the right.

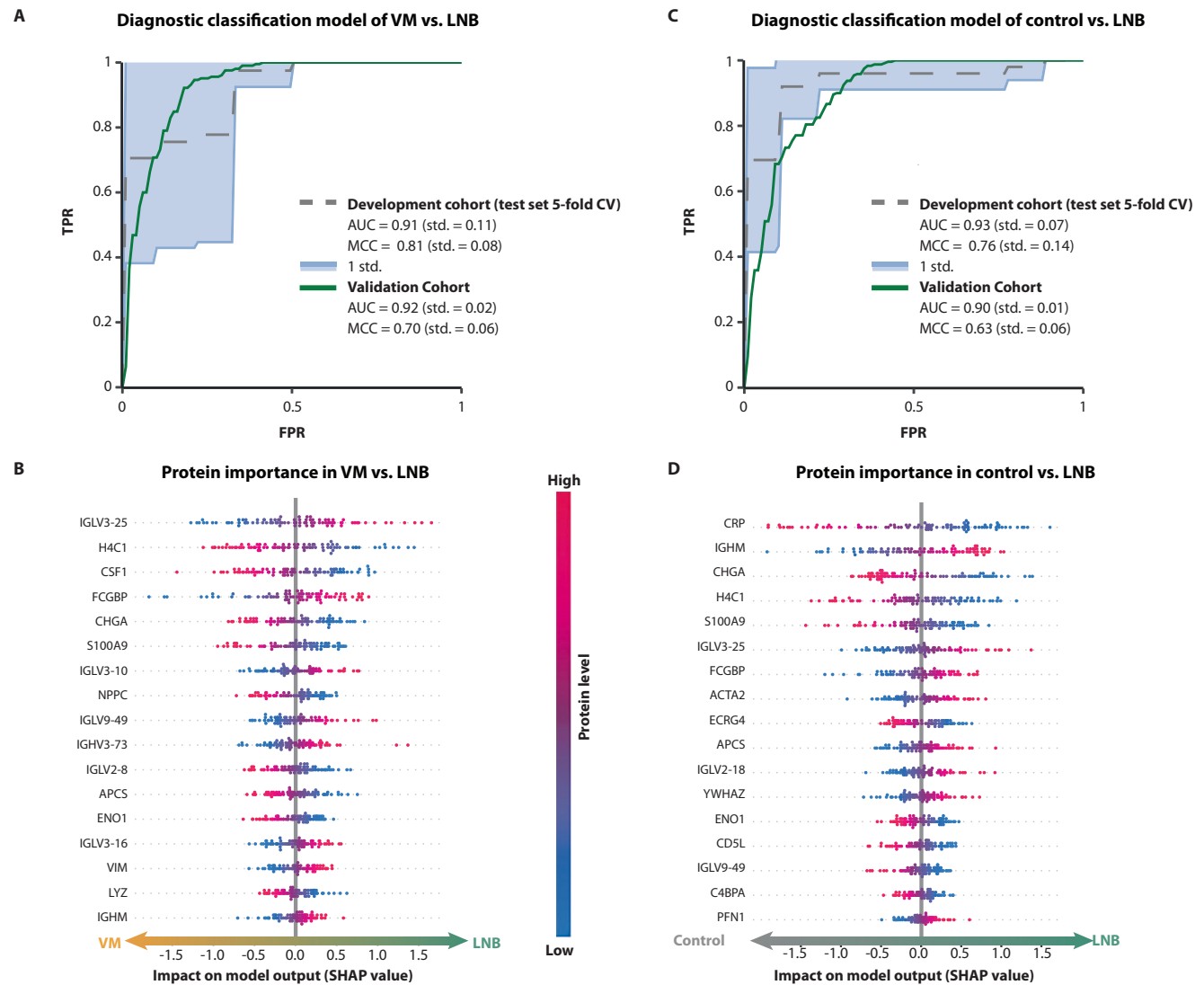

**Fig. 2 | Diagnostic classifier for Lyme Neuroborreliosis (LNB) based on machine learning and cerebrospinal fluid (CSF) proteomics. A** Area Under ROC Curve (AUC) for the final machine-learning model (Logistic Regression) developed to discriminate between LNB and viral meningitis (VM) cases in CSF samples. The dashed line represents the mean AUC on the test set from the 5-fold cross-validation (CV), semi-transparent error intervals are presented as mean values ± 1 standard deviation (std). The performance of the model on the validation cohort is presented with a green line. Prediction performance is presented with AUC and Matthews Correlation Coefficient (MCC) for each cohort. **B** Feature importance of

the LR model for the top 10 most predictive features depicted with SHAP values. **C** AUC for the final machine-learning model (Support Vector Classifier (SVC)) developed to discriminate between LNB and controls. The dashed line represents the mean AUC on the test set from the 5-fold cross-validation (CV), semi-transparent error intervals are presented as mean values ± 1 standard deviation (std). The performance of the model on the validation cohort is presented with a green line. **D** Feature importance of the SVC model for the top 10 most predictive features depicted with SHAP values. TPR true positive rate, FPR false positive rate.

(Fig. 2D), were similarly related to innate and humoral immune responses, neuroendocrine signaling, and cell damage.

### Plasma proteomics
**Plasma development cohort.** A total of 379 proteins were identified in plasma, with a median of 268 proteins quantified per sample (Supplementary Fig. 6). One sample was excluded due to low protein number. After filtering for missingness, a total of 232 proteins were included for further analysis.

A total of 68 proteins were significantly different between LNB and viral meningitis (Fig. 3A, Supplementary Table 2), and 69 for LNB versus controls (Fig. 3B). The overlap between these two comparisons of significant proteins was 36 (Fig. 3B, Supplementary Table 2).

**Plasma validation cohort.** In the validation cohort, a total of 25 proteins were significantly different in LNB compared to viral meningitis

and controls. Ten overlapped with the significant proteins from the development cohort (Fig. 3B and Supplementary Figs. 7 and 8).

**Protein signatures of LNB in plasma.** The differences in protein profiles between the three patient groups are dominated by proteins involved in innate and humoral immune responses (Fig. 3C). When comparing protein profiles of plasma samples from LNB and viral meningitis, patients with LNB generally seem to have a relative upregulation of proteins associated with innate immunity and complement activation (e.g., FCN3, SERPING1, SERPINA5), lipid metabolism (e.g., APOE, APOC1, APOM), and coagulation regulation (e.g., F13A1, PROC). In contrast, patients with viral meningitis have a plasma protein profile dominated by acute-phase and pro-inflammatory markers (e.g., CRP, S100A8, S100A9) and immunoglobulin production. In the comparison between LNB and controls, upregulated plasma-proteins in LNB are similarly involved in complement activation (e.g., C3, C5), coagulation

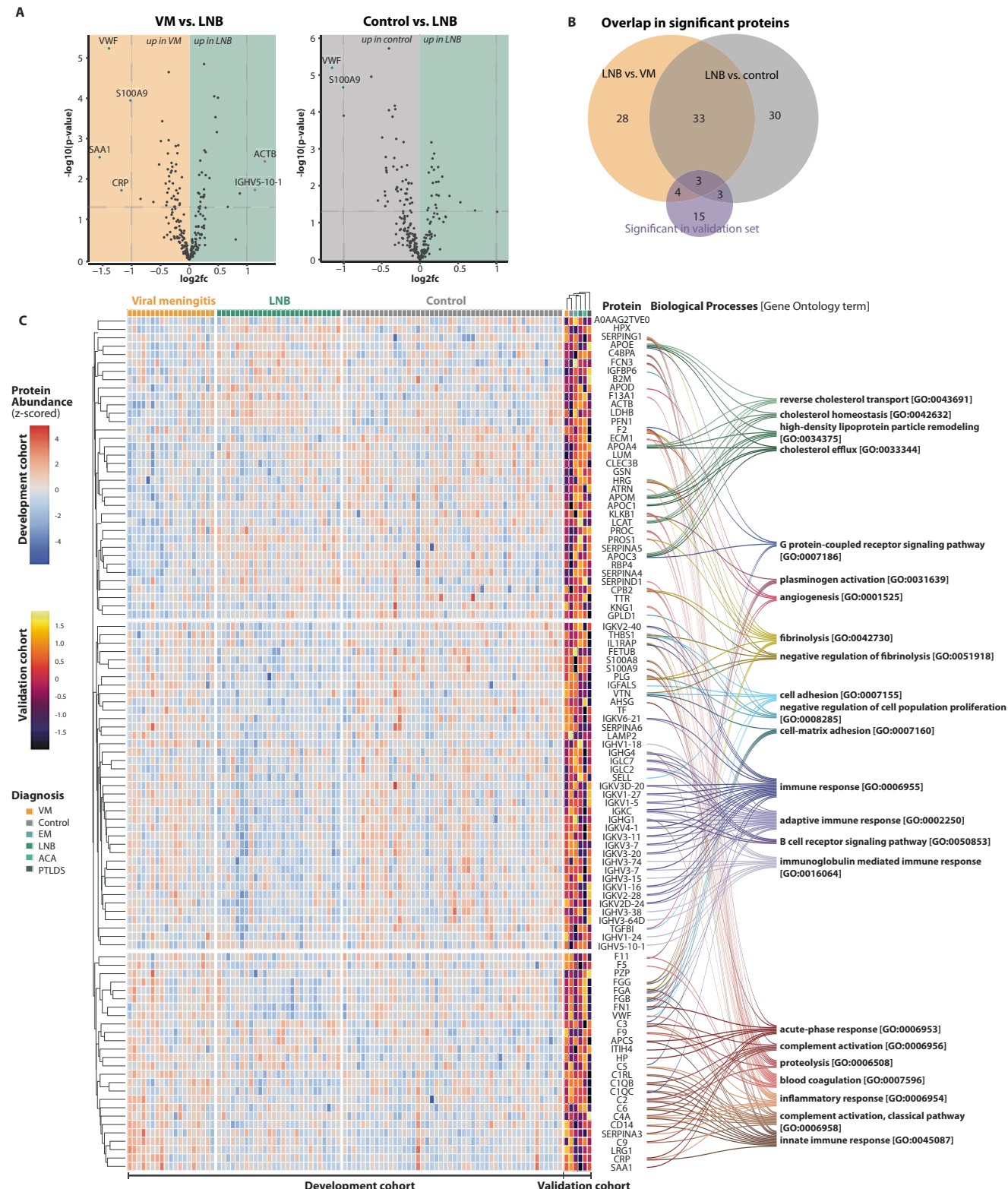

**Fig. 3 | Proteome changes in the plasma of Lyme Neuroborreliosis (LNB) patients. A** Volcano plots highlighting protein differences between Lyme neuroborreliosis (LNB) vs. viral meningitis (VM) and LNB vs. Controls in the development cohort in plasma samples. Significant proteins were identified by a two-tailed *t*-test adjusted for multiple hypothesis testing with Benjamini–Hochberg correction; adjusted *p* values < 0.05 were deemed statistically significant. Each point represents a protein, the color purple represents upregulated proteins with a log2 fold change larger than 1, the color blue represents downregulated proteins with a log2 fold smaller than −1, and the color black represents proteins with a log2 fold change between −1 and 1 or non-significant proteins. *X*-axis depicts log2 fold change, and *y*-axis depicts the log10 adjusted *p* value. Horizontal dashed lines represent the significance threshold after multiple hypothesis correction at *p* = 0.05. Vertical dashed lines represent large log2 fold changes: above 1 or below −1. **B** Overlap in significant proteins from the two volcano plots. **C** Heatmap of the significant proteins for each plasma sample in the development cohort and the corresponding mean from the validation cohort. The Gene Ontology term associated with the protein is depicted on the right.

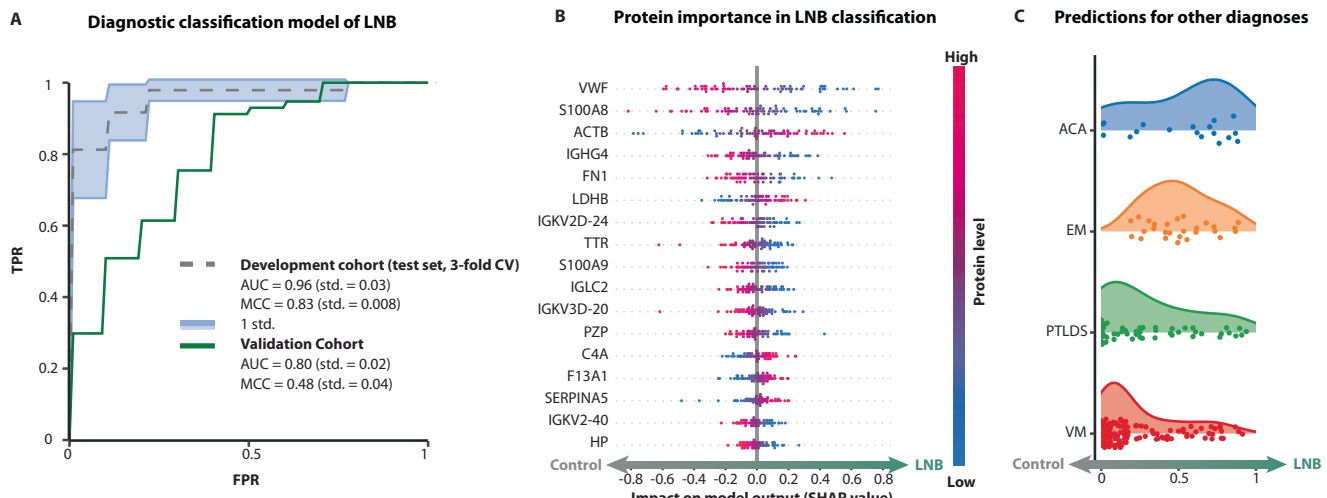

**Fig. 4 | Diagnostic classifier for Lyme Neuroborreliosis (LNB) based on machine learning and plasma proteomics. A** Area Under ROC Curve (AUC) for the final machine-learning (ML) model, Support Vector Classifier (SVC), developed to discriminate between Lyme neuroborreliosis (LNB) and controls in plasma samples. The dashed line represents the mean AUC on the test set from the 3-fold cross-validation (CV), semi-transparent error intervals are presented as mean values ± 1 standard deviation (std). The performance of the model on the validation cohort is presented with a green line. Prediction performance is presented with AUC and Matthews Correlation Coefficient (MCC) for each cohort. **B** Feature importance of the SVC model for the top 10 most predictive proteins depicted with SHAP values. **C** Distributions of predicted probabilities from the LNB classification model on other diagnostic groups. TPR true positive rate, FPR false positive rate, ACA acrodermatitis chronica atrophicans, EM erythema migrans, PTLDS post-treatment-Lyme-disease-syndrome, VM viral meningitis.

and inflammation regulation (e.g., SERPINA3, SERPIND1), and immune system modulators (e.g., HP, ITIH4). In contrast to CSF, most immunoglobulins had a higher protein level in the plasma of control individuals compared to LNB patients.

## Diagnostic classifier for LNB based on machine learning and plasma proteomics

Similar to the CSF analysis, plasma proteome data were subjected to a machine-learning algorithm and hyperparameter selection in a 3-fold cross-validation scheme. It was not deemed feasible to develop a model for the discrimination of viral meningitis and LNB due to the low number of viral meningitis cases; hence, the only model tested was the discrimination of controls and LNB. Of the twelve models tested, the SVC model obtained the best performance in the test set (Supplementary Fig. 9) with an AUC of 0.96 (std. = 0.03) and a MCC of 0.83 (std. = 0.008) (Fig. 4A). When applied on the validation cohort the model obtained an AUC of 0.80 (std. = 0.02) and a MCC of 0.48 (std. = 0.04) (Fig. 4A). According to the SHAP feature importance values, the most predictive proteins for the diagnostic classification of controls vs. LNB (Fig. 4B) were associated with innate immunity, humoral immune defense, coagulation and cellular metabolism. When the LNB classification model was applied to other diagnostic groups, it identified groups with active *Borrelia burgdorferi* s.l. infections; acrodermatitis chronica atrophicans and erythema migrans, as having significantly higher similarities to the LNB profile than groups with absence of active infection; Post-treatment-Lyme-disease-syndrome or unrelated etiology; viral meningitis. Mann-Whitney U Test (acrodermatitis chronica atrophicans + erythema migrans vs Post-treatment-Lyme-disease-syndrome + Viral meningitis): U statistic = 5475.0, *p* value = 2.65e-07 (Fig. 4C).

## Comparison of protein profiles

We found an overlap of 72 proteins that were significantly different between LNB and controls or viral meningitis in both CSF and plasma (Fig. 5A, Supplementary Table 3). The majority of overlapping proteins (62/72) were relatively upregulated in CSF from patients with LNB compared to controls and viral meningitis, and 30 of these were inversely regulated in plasma. This group primarily included

immunoglobulins (18/30) (Fig. 5B). We found four other studies investigating the proteome profiles of patients with LNB by Angel et al.[17], Gęgotek et al.[18], Fredriksson et al.[19], and Nilsson et al.[20]. The studies vary in sample sizes, technology, tissue, and disease focus (Fig. 5C, Supplementary Table 4). Two other studies applied untargeted mass-spectrometry, whereof Angel et al.[17] compared the CSF profiles of 26 cases with Lyme disease and 19 controls and reported findings overlapping with both CSF and plasma findings from this study. Another LC-MS-based study was performed by Gęgotek et al.[18] on serum from patients with LNB (*n* = 10) at different time points in order to evaluate the changes in proteome profiles before and after therapy. They also included a control group of 10 healthy individuals, where they found that 79 proteins significantly differed between LNB and controls. Also, here, the results overlapped with both CSF and plasma findings from this study. In the investigations performed by Fredriksson et al.[19] and Nilsson et al.[20], they applied targeted Olink protein profiling. Nilsson et al.[20] compared a subset of patients with PTLDS (*n* = 31) with patients with active LNB (*N* = 30) and identified 92 and 124 proteins with significant changes in serum and CSF, respectively. Several of the proteins they found were also found in our CSF analysis. Fredriksson et al.[19] analyzed serum from 119 pediatric patients with LNB (*n* = 61) and non-LNB (*n* = 58) and found a panel of 5 proteins that overlapped with results from the study by Nilsson, but no overlap was observed with the MS-based studies.

## Discussion

This exploratory study shows that mass-spectrometry-based proteomics combined with machine learning can identify distinct protein signatures in patients with LNB, enabling differentiation from patients with viral meningitis and non-LNB controls in CSF, and to a lesser extent in plasma. These findings provide new insights into the host response in LNB and should foster further research into the potential role of proteomics in LNB diagnosis and treatment monitoring.

To our knowledge, this is the first study exploring the potential use of untargeted proteomics in combination with machine learning as a diagnostic tool in a large and well-defined cohort of adult LNB patients, with evaluation of both CSF and plasma samples. Previous studies investigating the diagnostic potential of proteomics in LNB

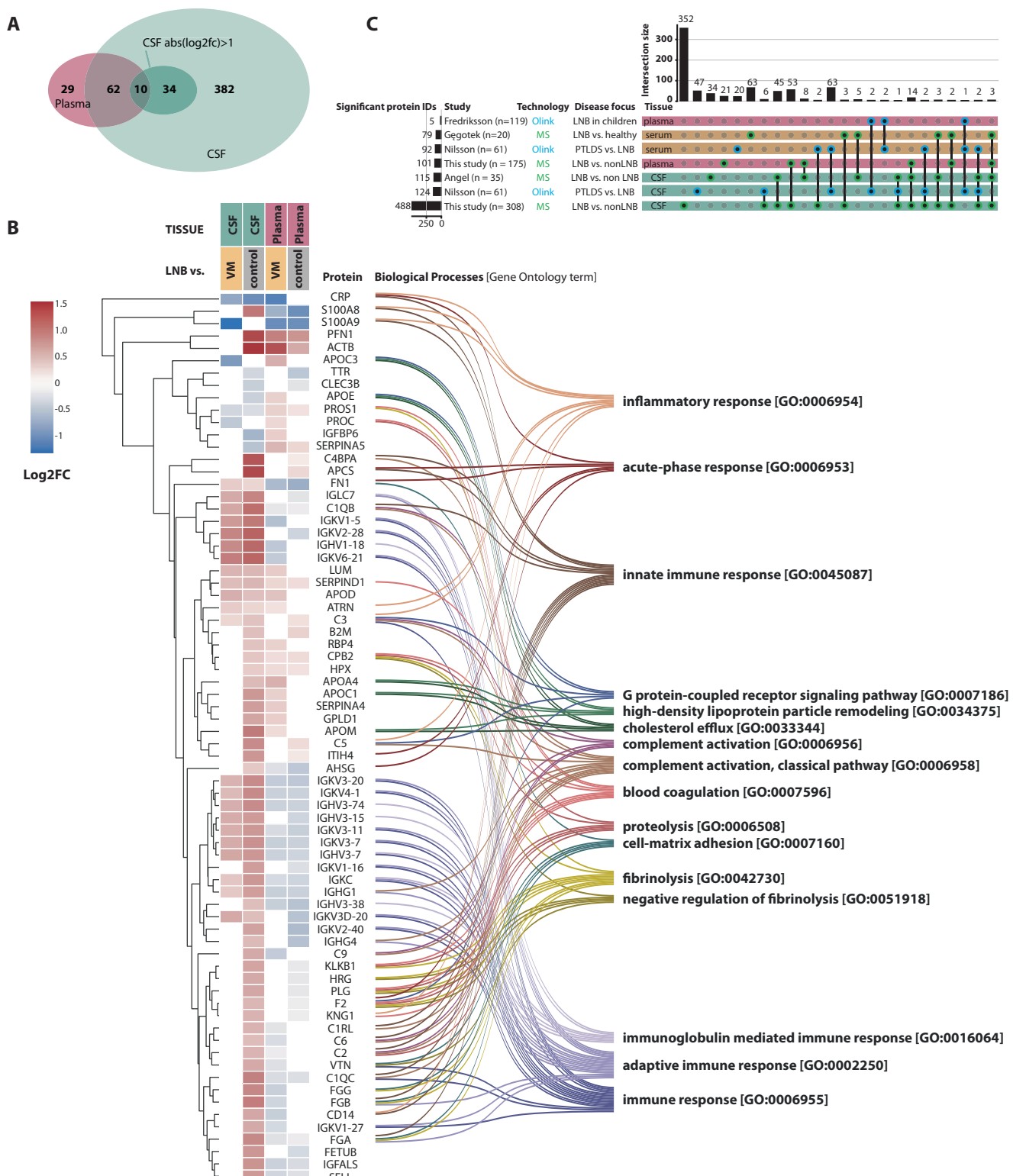

**Fig. 5 | Comparison of protein profiles. A** Venn diagram highlighting overlap between significant proteins found in this study in the plasma and cerebrospinal fluid (CSF) samples. Significant proteins were identified by a two-tailed *t*-test adjusted for multiple hypothesis testing with Benjamini−Hochberg correction; adjusted *p* values < 0.05 were deemed statistically significant. For the CSF findings, the abs(log2fc)>1 represents significant proteins, where the log2 fold change was higher than 1 or lower than −1. **B** Heatmap of the log2 fold changes between LNB and VM or controls of the 72 overlapping proteins. Empty fields represent non-significant findings for that comparison. Gene ontology terms for the proteins are summarized in the Sankey plot on the right. **C** Upset-plot highlighting the overlap between significant findings reported in this study and other studies.

have been limited by smaller sample sizes, methodology, and focus on other stages of *B. burgdorferi* s.l infection. Interestingly, Angel et al.[17] reported a diagnostic AUC of 0.8 on a panel of 13 proteins, where 3 (LYZ, C1QC, and FCGBP) were also among the significant proteins we found in CSF. Gęgotek et al.[18] and Nilsson et al.[20] did not report measures of diagnostic accuracy, but several proteins overlapped with our findings. Fredriksson et al.[19] found that a 5-protein-panel identified LNB with an AUC of 0.88 in children. Though the diagnostic groups and the use of blood samples were comparable to our study, it is unclear to what extent results in children can be extrapolated to adults. Overall, despite differences in study design, overlap of specific proteins was seen between studies and a consistent pattern of immune activation emerged across studies, emphasizing the role of innate immune responses, complement system, and humoral immune defense in the human host response to *B. burgdorferi* s.l. infection.

Our proteomic findings in CSF indicate a distinct immune response in LNB characterized by elevated levels of immunoglobulin chains, complement-related proteins, and proteins involved in immune cell migration and cytoskeletal dynamics. These findings suggest a highly targeted immune response against *B. burgdorferi* s.l. antigens in CSF, involving adaptive immunity with significant antibody production and complement activation. This aligns with established knowledge of host immune response during infections with *B. burgdorferi* s.l.[21]. It is worth noting that all patients with LNB in this study had a positive *B. burgdorferi* s.l.-specific intrathecal antibody index. Accordingly, the high levels of immunoglobulin proteins detected by proteomics are expected and may, in part, reflect this inclusion criterion. Whether a similar proteomic signature is detectable in patients with probable LNB, those with compatible clinical features and pleocytosis but lacking a positive antibody index, remains an open question that warrants further investigation in future studies.

Plasma and CSF both showed changes in immunoglobulin chains, complement factors, and innate immune modulators in LNB compared to controls, indicating a systemic and localized immune activation. The CSF findings highlight immunoglobulin diversity and a direct antibody response, reflecting the local immune reaction in the CNS. Meanwhile, plasma proteomics revealed a strong systemic immune engagement, with upregulation of innate immune components such as ficolins and coagulation proteins, underscoring systemic inflammation. Notably, viral meningitis exhibited a different pattern, with both plasma and CSF showing elevated acute-phase reactants (e.g., CRP, S100A9) and immunoglobulins, consistent with a strong inflammatory and humoral response. Opposing regulation of certain proteins in CSF and plasma in LNB likely reflects a compartmentalized immune strategy, where the CNS immune response is tightly regulated, while plasma changes mirror systemic inflammation. Additionally, factors such as blood-brain barrier permeability, local versus systemic protein production, and clearance mechanisms may further influence these patterns. Interestingly, we observe a higher relative abundance of proteins associated with the humoral immune response in plasma samples from VM and controls compared with LNB. These are likely caused by pathophysiological differences between the three groups. Thus, in LNB, the humoral response shows a clear compartmentalization to the CNS, whereas patients with VM may be more likely to present in the acute phase, where the immune response is more systemically distributed, and the group of non-LNB controls includes patients with Lyme borreliosis and other conditions with peripheral immune activation as the cause of their symptoms. These differences may explain why patients with VM and controls have higher relative abundances of immunoglobulins in plasma compared to LNB.

Additionally, structural proteins involved in immune cell motility were upregulated in CSF, indicating active immune cell recruitment within the CNS, specifically in LNB. Viral meningitis patients present a pattern more focused on inflammatory markers and metabolic activity, consistent with an acute viral immune response.

A small cluster of cytoskeleton and extracellular matrix proteins (VIM, ACTA2, PFN1, FN1, and ACTB) emerged among the top discriminatory markers, suggesting increased cell turnover in LNB. While expected in LNB vs. controls, this is more surprising when compared to viral meningitis. Notably, *B. burgdorferi* s.l. Spirochetes are hypothesized to extravasate early upon transmission and travel through the extracellular matrix to peripheral nerves, eventually reaching the borders of the CNS and creating the inflammatory basis of radiculoneuritis[22,23]. Vimentin (VIM), a class III intermediate filament that regulates myelination in axons of peripheral nerves[24], is particularly relevant in the context of LNB. This is due to our limited understanding of CNS entry by *B. burgdorferi* s.l. spirochetes, the frequent cranial nerve involvement in LNB with painful radiculoneuritis[9,25–27], and because proteins released from damaged nerves to the peripheral circulation could serve as early biomarkers. This has previously been shown with neurofilament light chain (NfL)[28]. Additionally, fibronectin (FN1), which is also found in this cluster, is considered essential for *B. burgdorferi* s.l. cell adhesion and migration through binding to *B. burgdorferi* s.l. surface proteins BBK32[29] and RevA[30].

The ability of proteomics to capture a broader range of host-response proteins, beyond the adaptive immune markers used in current assays, offers potential advantages for diagnosing early disease and for dynamic monitoring of disease development and treatment response, where antibody titers may remain elevated despite changes in disease activity.

These exploratory findings should encourage further investigation and external validation, including direct comparison of proteomics with existing diagnostic methods, to better assess potential clinical value. If validated in external cohorts, proteomics could, for example, aid in diagnosing LNB among adult patients with CNS infections with unknown etiology. This may be particularly relevant for PCR-negative suspected viral meningitis cases, where CSF proteomics suggestive of LNB could prompt further testing or the direct initiation of relevant antibacterial treatment. However, the most significant clinical implication would be the ability to diagnose and monitor LNB using a blood sample alone. It would spare patients the discomfort and risk for complications of lumbar puncture and reduce healthcare costs associated with referral to specialized healthcare facilities. A diagnostic blood test for LNB would likely reduce the diagnostic delay and thereby reduce the risk of residual symptoms. Even a modest improvement over current serology-based approaches could lead to fewer invasive procedures and significantly ease monitoring of treatment response and disease activity. Using the data reported in the ECDC systematic review[31] we calculate an MCC of 0.41 for the serologic assay, with PPV and NPV values of 0.31 and 0.97, respectively. In our dataset, an MCC of 0.48 corresponds to a PPV of 0.53 and an NPV of 0.93. While this perspective is encouraging, we emphasize that the performance was not compared directly to current serologic assays for blood-based diagnosis of LNB. This comparison remains to be addressed in future validation studies across diverse patient populations.

To further place these results in context, it is important to reflect on the methodological limitations and exploratory nature of the study. The retrospective design of this study made the establishment of cohorts dependent on available biobank samples, and the additional information available on included patients was limited. We were therefore not able to classify patients in full accordance with the European Federation of Neurological Societies diagnostic criteria[7], but only included patients with a first-time positive *B. burgdorferi* intrathecal antibody index as cases of LNB. Although this is a stringent and well-validated criterion for identifying true LNB patients with a very high degree of specificity[32–34], it excludes patients with early LNB who have not developed a positive intrathecal antibody index at the time of presentation. This will expectedly affect the generalizability of our models to such patients, and future studies should include patients

across the full clinical spectrum to explore these possibilities, including patients with other inflammatory CNS conditions (meningitis, encephalitis, CNS lymphoma, multiple sclerosis, etc.). Other factors such as disease stage, age, and sex can also affect the protein signatures and thereby the generalizability of our models[35–39]. For example, age differences between patient groups can act as a confounding factor in machine learning, as the model may learn age-related proteomic patterns instead of true disease-specific markers. This can artificially inflate performance metrics like AUC and reduce generalizability. However, our study population reflects real-world clinical variation, as patients were primarily recruited from referral hospitals. Furthermore, prioritizing clinically relevant diagnostic groups over strictly age- and sex-matched controls enhances the translational value of our findings. Thus, patients with viral meningitis were included to assess the ability of proteomics to differentiate LNB from other conditions with similar degrees of CNS inflammation. The control cohort consisted of individuals who were investigated for LNB due to clinical suspicion, as opposed to LNB vs a non-related control cohort.

Another methodological consideration has been to focus on diagnostic feasibility, more specifically, to employ a proteomic workflow with a short turnaround time suitable for clinical implementation. Achieving a sufficiently short turnaround time comes with the cost of a slightly reduced dynamic range, as prioritized in this study. This will inevitably affect the ability to detect very low-abundant proteins like chemokines and cytokines. Thus, the chemokine CXCL13, which is a well-validated biomarker of LNB in CSF, was not detected in this study, likely due to the limitations in mass-spectrometry range. Even without CXCL13 as part of the model, proteomics still achieved a high level of diagnostic accuracy for LNB.

In conclusion, our findings suggest that machine-learning-assisted proteomics could play a role in the development of novel diagnostic tools in LNB. More specifically, it could support diagnosis in cases where serology is ambiguous or delayed, such as early-stage disease or reinfection, and could offer a dynamic method for monitoring treatment response and disease activity in patients with persistent symptoms following treatment. A less invasive approach would also be particularly beneficial in children, who account for around 30% of all LNB patients, and often require general anesthesia to have a lumbar puncture performed[40–42]. However, these findings require further prospective validation to determine the true clinical value. This includes validation in external cohorts where geographic, ethnic, and disease prevalence differences may affect model performance, as well as longitudinal studies to assess the role of plasma proteomics in disease monitoring over time.

## Methods
### Study design
This study was an observational retrospective cohort study.

### Study population and setting
We identified a total of 483 CSF ($n = 308$) and plasma ($n = 175$) samples from adults (>18 years) with LNB, erythema migrans, acrodermatitis chronica atrophicans, post-treatment-Lyme-disease-syndrome, viral meningitis, and individuals who were investigated for suspected LNB, but had normal CSF (controls) from; (i) The Danish National Biobank (samples collected from 2001-2011)[43] and (ii) the Biobank of the Department of Infectious Diseases at Copenhagen University Hospital, Rigshospitalet (samples collected from 2016-2022). After informed patient consent for storage and future use in research was provided, CSF samples were labeled and stored at −80 °C, and blood samples in EDTA tubes were spun, and the supernatant was transferred to Eppendorf tubes, labeled, and stored at −80 °C. Samples from the two sites of origin were kept separate in the subsequent analyses to preserve independent cohorts for model development and validation.

### Cohorts
Samples were divided into four cohorts for diagnostic model development and validation in CSF and plasma, respectively. The CSF development cohort consisted of 145 CSF samples from patients diagnosed with LNB ($n = 49$), viral meningitis ($n = 44$), and controls ($n = 52$). The CSF validation cohort included a total of 163 CSF samples from patients diagnosed with LNB ($n = 69$), viral meningitis ($n = 45$), and controls ($n = 49$). The plasma development cohort included a total of 95 plasma samples from patients with LNB ($n = 27$), viral meningitis ($n = 20$) and controls ($n = 48$) whereas the plasma validation cohort consisted of 80 plasma samples from patients with LNB ($n = 10$), viral meningitis ($n = 18$), controls ($n = 20$) and an additional cohort of plasma samples from patients with manifestations of Lyme borreliosis without CNS involvement including post-treatment-Lyme-disease-syndrome ($n = 17$), erythema migrans ($n = 9$), and acrodermatitis chronica atrophicans ($n = 6$).

### Definitions of diagnostic groups
**LNB.** Patients diagnosed with LNB based on a first-time positive *B. burgdorferi*-specific intrathecal antibody test and the International Classification of Diseases 10th revision (ICD−10) diagnosis code Borreliosis: A69. The Borrelia-specific intrathecal antibody index was measured using a second-generation, flagella antigen-based capture enzyme immunoassay (IDEIA LNB, Oxoid, Hampshire, UK). The test result was expressed as a specific capture optical density (OD) value in CSF/OD-serum x (OD-CSF- OD-serum). An elevated (positive) index was assumed if the specific capture IgG and/or IgM index was >0.3 in combination with OD-CSF ≥ 0.15. CSF leukocyte counts were available from 39/155 patients with LNB (mean of $155 \times 10^6$ cells/L).

**Viral meningitis.** Patients with a final clinical diagnosis of viral meningitis were determined by an infectious disease specialist at a tertiary referral university hospital. The diagnosis was based on clinical presentation, CSF pleocytosis, and exclusion of other diagnoses. CSF samples were tested with BioFire® Meningitis/Encephalitis (ME) Panel (BioFire Diagnostics, LLC, 515 Colorow Drive, Salt Lake City, UT 84108, USA) and supplementary in-house specific HSV and VZV PCR on clinical indication.

In 76 cases, a PCR-verified viral etiology was determined (Herpes simplex virus 2 ($n = 32$), Enterovirus ($n = 30$), Varicella zoster ($n = 9$), Influenza A virus ($n = 3$), EBV ($n = 1$), Toscana virus ($n = 1$)). CSF leukocyte counts were available from 22/127 patients with VM (mean of $172 \times 10^6$ cells/L).

**Controls.** Individuals who had a lumbar puncture performed because LNB was suspected, but where white-blood-cell (WBC) count in the CSF was within normal reference ranges (CSF-WBC: $0-5 \times 10^6$/L) and the *B. burgdorferi* s.l. specific intrathecal antibody index was negative. Information on final diagnoses was available for 122/169 of the control samples. The diagnostic groups were: Lyme borreliosis (erythema migrans, and post-treatment-Lyme-disease-syndrome) ($n = 26$), other infectious diseases (Epstein–Barr virus (EBV), COVID19, influenza, sinusitis, tonsilitis, rickettsiosis, and presumed viral infection without specified etiology) ($n = 28$), non-infectious inflammatory conditions (rheumatoid arthritis, temporal arteritis, monoclonal gammopathy of undetermined significance (MGUS) and B-cell lymphoma) ($n = 4$), neurological conditions (Bell's palsy, Parkinson's disease, migraine, tension headache and peripheral neuropathy) ($n = 16$), degenerative condition of the spine (spinal stenosis, spinal disc herniation) ($n = 5$). In the remaining 43 cases, no final diagnostic conclusion was reached.

**Lyme borreliosis group.** Plasma samples from patients with different presentations of Lyme borreliosis, including Post-treatment-Lyme-disease-syndrome, erythema migrans, and acrodermatitis chronica

atrophicans. The definitions of these clinical Lyme borreliosis manifestations were in line with published European guidelines and case definitions[7,44–47]. All patients in this cohort were evaluated at the Unit for Tick-Borne Infections, which is an outpatient clinic under the Department of Infectious Diseases at Copenhagen University Hospital, Rigshospitalet. Patients were assessed by two experienced infectious diseases specialists[48].

**Post-treatment-Lyme-disease-syndrome.** If symptoms related to a previously verified and treated Lyme borreliosis manifestation persisted for more than 6 months, the condition was termed post-treatment Lyme disease syndrome[7,47,49].

**Erythema migrans.** We defined erythema migrans as early localized Lyme borreliosis, solely from the clinical presentation[46,47].

**Acrodermatitis chronica atrophicans.** Clinical presentation compatible with acrodermatitis chronica atrophicans and elevated serum *B. burgdorferi* sl if possible, a skin sample was obtained for histological examination to further confirm the diagnosis by Borrelia PCR[46].

### Proteomic analysis
CSF and plasma samples were thawed, and 100 μl were transferred to 96-well plates for analysis. The sample preparation was optimized based on the previously published methods described in refs. 50,51. Twenty microliters of CSF were first denatured for 10 min with 30 μl PreOmics Lysis buffer, while 5 μl plasma was denatured for 10 min with 45 μl PreOmics Lysis buffer[52]. Both sample types were subsequently digested for 4 h using LysC/trypsin enzyme mix. The resulting peptides were purified using two-gauge SDB-RPS StageTips (approximately 20 min), and the eluate was analyzed on an Evosep One (Evosep Biosystem, Denmark) liquid chromatography system, coupled online to an Orbitrap Exploris 480 mass spectrometer. Data acquisition was performed in data-independent analysis (DIA), using 60 samples per day gradient and an 8 cm Pepsep column. Mass spectrometry was performed in positive ionization mode using data-dependent tMSn scanning, fragmentation was carried out by higher-energy collisional dissociation (HCD) with a fixed, normalized collision energy of 30% and spectra were acquired at a resolution of 30,000 (Orbitrap), with a maximum injection time of 28 ms.

### Data processing
Initial data processing of the mass-spectrometry raw files was performed with DIA-NN version 1.9 in a data-independent search[53].

The DIA-NN data underwent further processing using the Clinical Knowledge Graph, alphapeptstats, and Jupyter Notebook[54,55]. Initially, a stringent filter for missing data was applied: (1) samples with low protein count, defined by a value below 1.5IQR from the 25th quantile of the combined distribution, were excluded, and (2) proteins with a missingness of more than 40% across samples were excluded. Data was log2 transformed. The remaining missing values were imputed with a variational autoencoder using the PIMMS software[56]. Assessment of sample quality was conducted as previously described[56]. Batch correction was executed using Combat to overcome potential plate-specific bias on subsequent analyses[57].

### Data analysis
Proteins exhibiting significantly different levels between the cohorts were identified by unpaired *t*-tests. Multiple hypothesis correction was applied with the Benjamini–Hochberg method, with adjusted *P* values < 0.05 deemed statistically significant. *P* values and protein abundances were visualized in Volcano plots, with -log10(corrected *p* value) and log2 of protein fold change between groups. Venn diagrams were used to visualize overlaps in significant proteins between statistical comparisons. Heatmaps in combination with Sankey plots of gene ontology terms were used to highlight protein changes of the significant proteins together with their biological processes[58]. In the heatmaps, protein abundances were z-scored. Gene ontology terms were retrieved for each protein through UNIPROT. Gene ontology terms with a frequency of less than 5% across proteins were not visualized.

### Model development
Z-scored data from the development cohorts were analyzed using supervised machine learning to explore the potential for a diagnostic signature for LNB. Significant proteins identified by *t*-test analysis on the development cohorts were used as input features in the training data. Our data sets, comprising both samples from patients diagnosed with LNB and viral meningitis or control samples, were partitioned into a training set for model development and a test set for model validation using a 5-fold or 3-fold cross-validation (CV) approach for CSF and plasma, respectively. The number of CV-folds was decided based on sample size, with a minimum of 30 samples in each set. Both sets maintained an equal ratio of positive and negative cases (stratified k-fold cross-validation). The classification target used for analysis was the diagnosis of LNB and viral meningitis or LNB and control (yes/no). Twelve different machine-learning algorithms from the scikit-learn Python library were tested to determine which algorithm modeled the data best. For each algorithm, optimal hyperparameters and features were determined during cross-validation based on the weighted F1 score. The most appropriate machine-learning model was determined by testing the performance of all twelve fitted algorithms based on the AUC and MCC in the development test set cross-validation. Model calibration was subsequently assessed with the CalibratedClassifierCV integrated in Scikit-learn. Calibration was only applied if it improved the model performance. Feature importance was highlighted for the top 10 most predictive features with SHAP values[59].

### Model validation
The models trained on the development cohorts were subsequently applied to the validation cohorts, and the performance was assessed by AUC and MCC. ROC curves and confusion matrices were used to visualize the performance of the classifiers.

### Statistics and reproducibility
Sample sizes were determined by the availability of eligible CSF and plasma samples from the participating biobanks within the defined study period; thus, no statistical method was used to predetermine sample size. No data were excluded from the analyses after the quality control filtering described above. The experiments were not randomized. The investigators were not blinded to allocation during experiments and outcome assessment.

Statistical analyses were conducted in Python using scikit-learn and associated scientific computing libraries. Differences in protein abundances between groups were assessed using unpaired *t*-tests with Benjamini–Hochberg correction for multiple testing, and adjusted *P* values < 0.05 were considered significant. Machine-learning model development and evaluation used stratified k-fold cross-validation (5-fold for CSF, 3-fold for plasma), maintaining balanced positive and negative case ratios across folds. Model performance was evaluated using the area under the receiver operating characteristic curve (AUC) and the MCC.

This retrospective observational study used independent discovery and validation cohorts to support reproducibility. All raw mass-spectrometry data and processed protein quantifications are publicly available to enable independent reanalysis.

### Ethics and inclusion statement
This study was conducted in Denmark under appropriate ethical approvals, including approval from the Knowledge Center for Data

Reviews (P-2019-707) and the local ethics committee (H–17024315 and H–19027317). The biobanks were approved by the Danish Data Protection Agency (Rigshospitalet: j.nr.: 2012-41-0036). Samples were obtained during diagnostic investigation, and all patients had given their informed consent to the storage of biological material and its future use in research. The patients did not receive any compensation for their inclusion. The study was conducted within an established clinical research infrastructure, without transfer of biological materials across borders. Our study cohorts include individuals across a broad age range and diverse clinical backgrounds, reflecting real-world patient populations encountered in a tertiary care setting. The study addresses a clinically relevant diagnostic challenge in the local healthcare system, and future work will aim to validate these findings in more diverse clinical and international cohorts. The research team includes local clinical and scientific experts who contributed to all stages of the study, from design and data analysis to manuscript authorship. Roles and responsibilities were agreed upon in advance.

## Data availability

The proteomics data generated in this study have been deposited to the ProteomeXchange Consortium via the Proteomics Identifications Database (PRIDE) partner repository[60] with the identifiers PXD067474 and PXD067583 for CSF and plasma, respectively. The processed proteomics data are available at GitHub via https://github.com/WewerAlbrechtsenLab/LymeNeuroborreliosis together with the Source data files.

## Code availability

All code and data necessary to reproduce the analysis and figures associated with this publication are available at GitHub via https://github.com/WewerAlbrechtsenLab/LymeNeuroborreliosis.

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

## Acknowledgements

The authors express their gratitude to Christine Rasmussen from the Department of Clinical Biochemistry at Copenhagen University Hospital, Bispebjerg, for her dedicated efforts in planning, preparing, and conducting the proteomic analysis. Additionally, the authors extend their appreciation to the Clinical Proteomic Group at the NNF Center for Protein Research, University of Copenhagen, for sharing their expertise on mass spectrometry. The authors would also like to thank lab technicians Stine Østergaard and Dorthe Hass from the clinical research unit at the Department of Infectious Diseases, Rigshospitalet, for their great help with handling and sending samples for analysis. The authors further acknowledge the financial support that enabled this study: NNF21SA0 072746, NNF23OC0084970, NNF19OC0055001, NNF24OC0088402, and 1052-00003B supported N.J.W.A. NNF14CC0001 supported the Novo Nordisk Foundation Center for Protein Research and thereby A.B.N., L.D., and M.M. Grant E-23778-06 supported L.F. Grant R366-2021–127 supported A.M.L.

## Author contributions

N.J.W.A. and A.M.L. jointly conceived the study. L.F., H.M., A.J.H. and L.H.H. collected the samples. C.R. prepared the samples for analysis, and L.D. performed the mass-spectrometry measurements and raw data processing. A.B.N. conducted the data analysis, and together with L.F., L.D., M.E.O., M.M., H.M., A.M.L. and N.J.W.A. performed the final data analysis and interpretation. All authors contributed with intellectual

input and critical discussion. A.B.N. and L.F. wrote the initial draft. All authors reviewed and approved the final version of the manuscript.

## Competing interests

A.M.L. reports speakers' honorarium/travel grants/advisory board activity and unrestricted grant from Gilead, speakers' honorarium/travel grants from GSK, speaker's honorarium/advisory board activity from Pfizer outside this work. N.J.W.A. has received funding, served on scientific advisory panels, and/or speakers bureaus for Boehringer Ingelheim, MSD/Merck, Novo Nordisk, EvoSep, ROCHE, Janssen, and Mercodia. A.J.H. reports a research collaboration agreement with Pfizer unrelated to this work. M.M. is an indirect shareholder in Evosep Biosystems. None of the other authors reports any conflict of interest. The funding agencies, the Novo Nordisk Foundation, the Independent Research Fund Denmark, the Research Fund of Rigshospitalet, and the Lundbeck Foundation had no role in the design and conduct of the study; collection, management, analyses, and interpretation of the data; preparation, review, or approval of the manuscript; and decision to submit the manuscript for publication.
