## [Transparent Peer Review file · Nature Communications]

The diagnostic potential of proteomics and machine learning in Lyme neuroborreliosis

Corresponding Author: Dr Nicolai Wewer Albrechtsen

Version 0:

Reviewer comments:

Reviewer #1

(Remarks to the Author)
Comments to the authors

The manuscript presented by Bach Nielsen and co-authors investigates machine learning-assisted MS-based proteomics for a rapid and accurate diagnosis of Lyme neuroborreliosis. Any tool, any new method that will improve the diagnosis of this nervous system infection is very valuable in the field of Lyme disease. In particular, the possibility of making a diagnosis from blood, in a less invasive way than a lumbar puncture, is very promising. One of the strengths of this work is that it was based on a large number of samples (483 CSF and 175 plasma) with development and validation cohorts. The analytical workflow is robust and the analytical equipments and softwares correspond to the state of the art for non-targeted proteomics. The manuscript is well written, the patients and methods thoroughly described and I have only some minor remarks.

Lines 142-143 and Figure 1: In the comparison control vs LNB, 41 proteins are significantly expressed with a fold change larger than 2. Quote Figure A (Volcano plot) and not Figure B (Overlap).

While the 10 proteins from the VM vs LNB comparison are visible on Figure 1B, 43 proteins (33+7+3) are recorded on Figure 1B for control vs LNB. What makes the difference between 41 and 43?

Line 145: This sentence refers to Figure 1B, not Figure 1C

Line 202: The number of proteins identified in the plasma samples (median of 268 proteins) is quite low, which is not surprising in the absence of depletion or prefractionation, but only allows the more abundant proteins to be detected. The authors chose to use a very short 8 cm column (probably to increase the number of samples per day). What impact did this length of column have on the number of proteins identified, compared with previous plasma studies in the literature using much longer columns?

Figure 3A: To be consistent note "Control vs LNB" on the Volcano plot and not "LNB control vs LNB".

Figure 3B: Line 205 states that 61 proteins are significantly different in VM vs LNB. However, on Figure 3B, a total of 68 proteins (28+33+4+3) are in the corresponding circles. Same remark for the 63 proteins stated Line 206 (LNB vs control), while a total of 69 proteins (33+30+3+3) are in the corresponding circles. What make these differences?

Line 194 (Fig 3B legend): As seen on Figure 3A and Supplemental Table 2, only 2 proteins (VWF, S100A9) exhibit a fold change larger than 2 in LNB vs control. Similar remark for VM vs LNB with only 6 proteins with a fold change larger than 2. There must be an error in the, since all the significantly proteins are represented, and not only the significant proteins with a fold change larger than 2.

Line 207: This sentence refers to Figure 3B, not Figure 3C

Given that, for plasma, all the significant proteins have been taken into account (and not just those that vary with a fold change larger than 2 as in CSF), are there any consequences for the robustness of a blood-based clinical test?

Line 636 (Mat & Meth): As the MS equipment and the column are different from those used in the cited publications^{44,45,46}, the authors should give more details on LC conditions (e.g. gradient duration) and MS acquisition parameters (or cite additional publications).

(Remarks on code availability)

Reviewer #2

(Remarks to the Author)

The study explores the potential of mass spectrometry-based proteomics as a novel diagnostic tool for Lyme neuroborreliosis (LNB) in CSF and plasma of adults. The authors' stated aim is to develop a test for early diagnosis of LNB, avoiding the need for a lumbar puncture.

They analyzed stored CSF and plasma samples from adult patients with LNB, viral meningitis, controls, and other Lyme borreliosis manifestations and used machine learning analyses to identify panels of proteins that could distinguish between groups.

There are issues with this work. The first is the definition of the samples, in particular LNB and controls. LNB was defined based on a first-time positive *B. burgdorferi*-specific intrathecal antibody test and the ICD-10 code A69 for Borreliosis (which includes all presentations). There is no other information on the clinical presentation or information on pleocytosis.

The controls were individuals who had no pleocytosis and a negative *B. burgdorferi* specific intrathecal antibody index after being investigated for suspected LNB. There is no information if these patients had another diagnosis (of Lyme borreliosis or other disease). It is unclear if these controls would not include patients with other presentations of Lyme borreliosis.

The requirement of a positive IT antibody test anchors the LNB group to this definition. This is reflected on the list of differentially expressed proteins with an absolute log₂ fold change larger than 1 provided in in supplementary table 1. I rerun the list using online available tools, and found that 5 of 7 upregulated proteins in LNB compared with viral meningitis (VM) are within the GO CC term immunoglobulin complex GO:0019814, one to IgA immunoglobulin complex, and the other is Fc gamma binding protein. Similar findings are seen compared with controls, where most proteins are related to immunoglobulin complex and related terms. Also shown on the "signature of LNB" where "immunoglobulins were generally upregulated, while proteins involved in innate immune responses and neuroendocrine signaling were relatively downregulated in LNB compared to viral meningitis and controls". These issues are not discussed in the manuscript.

The requirement of positive index excludes patients with possible LNB by EFNS criteria who present with pleocytosis and typical clinical presentation. Studies have shown that these patients have a shorter duration of symptoms compared with patients with positive IT index, therefore representing an earlier phase of illness.

The fact that CXCL13 was not found to be increased in CSF of LNB patients, or to be part of the models is highly problematic. While the explanation that "the limited detection range inherent to this approach means that certain low-abundance but highly disease-specific proteins, such as CXCL13, may not be captured. Given that previous proteomic studies in LNB have also failed to detect CXCL13, this highlights a general challenge in proteomic biomarker discovery rather than a limitation of our study" is possible, it does not recommend the methodology for discovery. CXCL13 is measured in higher levels in CSF in acute LNB than in other neuroinflammatory diseases and may precede antibody production. It has been proposed as both an early diagnostic marker and a marker of treatment response.

In terms of the ML, twelve models were tested for the CSF data and for plasma data. A LR model was chosen for LNB vs viral meningitis, and an SVC model for LNB vs. controls. No models were developed using the 3 groups. The LR model for LNB vs VM had a MCC of 0.81 on the test set and 0.7 on the validation. For LNB vs controls, the SVC model had an MCC of 0.76 on the test set and 0.63 on the validation. While these are good levels of accuracy, I would disagree with the statement that "Overall, MS-based proteomics coupled with ML demonstrated strong potential as a diagnostic tool for LNB, with particular proteins playing key roles in distinguishing between conditions." As the LNB group was defined by positive IT index, this would have a MCC of 1. It is also highly likely that Lyme serological test would also have a higher MCC, and possibly CXCL13.

The plasma analysis showed that VM patients and controls had higher immunoglobulin levels than LNB patients. These are highly unexpected and unexplained results, particularly comparing with controls. An explanation for these findings, including technical issues need to be provided.

The ML model for plasma focused only in differentiating between controls and LNB. The SVC model chosen had a MCC of 0.83 on the test set, but only 0.48 in the validation set.

The authors stated that the model identified acrodermatitis chronica atrophicans and erythema migrans compared to post-treatment Lyme disease syndrome and viral meningitis. However, little detailed information is given regarding the individual presentations, the numbers are small for some, and no comparison is done to available methodologies

Another difficulty is the possible variation caused by age differences in the groups. While there is no analysis of the differences, by the information provided (median age and IQR) is possible that there are significant differences between the groups. This is mentioned only in brief in the discussion.

The discussion highly overrepresents the results, and downplays the problems and limitations of the study. Examples include statements like "MS-based proteomics can accurately differentiate patients with LNB from patients with viral meningitis and non-LNB controls", "observed a high level of diagnostic accuracy in plasma, highlighting the potential of blood-based proteomics as a clinically valuable tool" and "In conclusion, our findings demonstrate that MS-based proteomics can accurately distinguish LNB from controls in both CSF and plasma". The results do not support these statements. It is likely that the models would not perform any better than current antibody-based assays.

Figures are difficult to read due to small print, particularly figures 1 and 3. There are label overlaps in Figure 1 and supplementary figures 2 and 3.

(Remarks on code availability)

Reviewer #3

(Remarks to the Author)

The manuscript by Nielsen et al reports the use of an MS-based proteomics technique for Lyme neuroborreliosis in adults. This is quite an interesting paper expanding on the value of host-response profiles for predicting disease. I have some comments/questions for consideration:

(1) The authors should be praised for having a diverse sample set including non-Lyme diseases. For the viral meningitis cases, what assay was used for pathogen detection. Should state the assay. For example if BioFire FilmArray ME Panel was used, the HSV target has some limitations. In clinical practice, HSV testing may need to be tested on a separate assay, due to concerns on assay performance.

(2) On the same topic, the authors noted that a lot of the signal detected MS in CSF were immunoglobulins. Can the authors suggest, perhaps in the Discussion Section, how other diseases such as multiple sclerosis which has unique immunoglobulin profiles in CSF, would impact the assay performance?

(3) Can the authors comment on how long the test takes? This helps discuss how translatable it is for real world practice if this is ever adopted for routine clinical testing. I see a comment of 2 hours in line 90. Is this true, in the context of total testing time, not just the mass spec phase and/or informatics. What is the turnaround time from sample to fully verified result?

(4) For machine learning (ML), how did the authors select the 12 models? At least logistic regression (LR) and support vector machine (SVM) was used. Was the use of LR and SVM based on a priori experience, familiarity, or other? The reason is there's been some studies showing an agnostic approach using a range of ML techniques could identify better models. Automated ML software has this advantage, although, it won't replace data scientists, some studies using MALDI-TOF-MS for infectious disease host responses found automated ML systems finding better models compared.

(5) Often, ML models should be further validated with a tertiary dataset, which helps with generalization. Can the authors comment on how the results from this initial study are likely to be translated to samples from another location? For instance, disease prevalence is a limitation here as the authors noted for viral meningitis. If a new dataset were to have a higher prevalence of different diseases, could this alter the results?

(6) For ML models, are there any other variables used to evaluate the model such as F-statistic and/or Brier Scores?

(Remarks on code availability)

Version 1:

Reviewer comments:

Reviewer #2

(Remarks to the Author)

REVIEWER COMMENTS

This is the 2nd review of this study, which explores the potential of mass spectrometry-based proteomics as a novel diagnostic tool for Lyme neuroborreliosis (LNB) in CSF and plasma of adults. Again, the stated aim is to develop a test for early diagnosis of LNB, avoiding the need for a lumbar puncture.

The issues raised on my previous review are still unresolved. The main one is that as written, the data does not support the claims. This starts with the title, which claims a rapid and accurate diagnosis of Lyme neuroborreliosis in adults using mass-spectrometry-based proteomics and continues in the abstract and the discussion. The major issue is that this approach is unlikely to be any better than current conventional serology.

This is shown by the results of the diagnostic classifier in plasma, where the model was only able to differentiate LNB patients from controls (and that with AUC of 0.80 and MCC of 0.48), performing much worse than what is expected for serologic assays. The models were no able to differentiate LNB from other LB patients (EM, ACA). This is not different from current antibody-based tests.

To say that these results support a "diagnostic blood-based assays for LNB" and "have clinical relevance" is an overstatement. It is unclear to me how the investigators can claim improvement in diagnostic using this more expensive and resource intensive approach, or that this could be a valuable clinical tool when there is no evidence that this offers any benefit comparing with current methods. Their answer is that this would be addressed in a prospective study, but this is not true. This analysis can easily be done using current antibody assays on the samples tested by proteomics, as well as using laboratory data that already exists for all the patients and controls, as I am sure all have had Lyme disease serology

performed as part of the evaluation. This comparison should be added to this study.

Most of the discussion should be rewritten as it continues to overrepresents the results and downplays the problems and limitations of the study. Again, the current data does not support that this approach is a clinically valuable tool, or that it could be used for initial diagnosis and monitoring of treatment response, or that could distinguish Lyme neuroborreliosis any better than current assays (which are much cheaper and faster). While there are many parts of the discussion that need to be modified, in particular, lines 284-289, 370-384, 397-400

It would be best if the authors focused on the exploratory aspects of the research.

An issue that has not been answered is that the findings of many increased protein related to immunoglobulins in Lyme neuroborreliosis CSF are likely to due to the requirement of a positive IT antibody test for LNB. The answer provided and the text added to the manuscript do not address this core issue and reflect a lack of understanding of the definition bias. The only way to ascertain the effect of the positive IT an index requirement in the definition would be to compare the findings on probable LNB patients who had clinical features and pleocytosis but not positive IT ab index. This potential bias should be discussed.

While the authors added that they were "likely to have missed some patients with early LNB" (line 394), a more detailed discussion of this issue should be added. While the positive Bb IT antibody index is likely a validated identifier of LNB in the correct setting, it is a later finding, as it has been shown to correlate with duration of illness. As the stated aim is for an early diagnostic test, the lack of this group (clinical picture + pleocytosis but negative IT) and the implications for the study should be more clearly stated.

As the study does not have patients with meningitis, encephalitis, multiple sclerosis and only one control with CNS lymphoma, there is no data to support the added text (lines 397-400) "Thus, the demonstrated ability of proteomics to differentiate LNB from VM is important for the applicability of the test in settings where other inflammatory CNS conditions are differential diagnoses (meningitis, encephalitis, CNS lymphoma, multiple sclerosis etc.)"

I also take issue with some of the author's response to reviewer 3.

2-While typical acute LNB and multiple sclerosis have mostly distinct clinical features, the aim would be for a general test and false positive results would have major impact on clinical utility.

3-I am not sure what is the basis for the author's statement that "even without further optimizations, we believe this level of throughput already represents a significant improvement over current clinical methods, which often require days to weeks, and demonstrates promising potential for future clinical implementation." Current assays for antibody-based diagnosis of Lyme borreliosis DO NOT require days to weeks as stated by the authors. These are EIA-based tests that can be rapidly performed once the sample was received. Even immunoblots can be performed much faster than proteomics. Are the authors adding the transport time to a central laboratory in their assumption and/or time due to test batching?

(Remarks on code availability)

Reviewer #3

(Remarks to the Author)

Thank you for addressing the questions/comments. I feel the authors have adequately addressed them and I have nothing further to add.

(Remarks on code availability)

Version 2:

Reviewer comments:

Reviewer #2

(Remarks to the Author)

This is the 3rd review of this study, which explores the potential of mass spectrometry-based proteomics as a novel diagnostic tool for Lyme neuroborreliosis (LNB) in CSF and plasma of adults. The manuscript is improved, but there are still points to be addressed.

My comments:

Abstract: while Lyme neuroborreliosis can be severe, most cases are not (when compared with bacterial or fungal meningitis). I would remove the adjective.

Few patients with LNB will require "continuous monitoring of disease activity". I would remove this from the abstract

Introduction:

Lines 74-75. Measurement of intrathecal antibody production for diagnosis of LNB is a requirement only in Europe. This is not required in the US. This need to be clarified in the text. There should also be a discussion of the differences in CSF testing between US and Europe, as a positive intrathecal antibody index is thought to be less common in the US.

Discussion:

Line 282-286. I am confused about the comments on the study by Angel et al. What do the authors mean by:

"However, the focus on other Lyme disease manifestations rather than LNB likely reflects the, somewhat debated, geographical difference in prevalence of neurological involvement with a B. burgdorferi s.l. strain-dependent lower incidence in North America compared to Europe"^{18,19}.

The sentence as written does not make sense. It appears to be truncated. The study only included LNB patients (as all had CSF pleocytosis but 1 patient that had signs of CNS involvement), so the comment regarding other manifestations is unclear.

The citations in the text are confusing. I would prefer that the reference number comes soon after the citation, and not at the end of the sentences. I am also confused by the information in parenthesis. Is it necessary to say Angel et al. (2013), Fredriksson et al. (Sweden, 2024), Gegotek et al. (Poland, 2024)?

I find the discussions of these previous studies underwhelming. I would expect at least a discussion of the comparison of the data from this study to the other studies in LNB, particularly references 17, 20 and the cohort with acute LNB from reference 22. These comparisons can be provided in detailed as supplemental information.

Also, a recent publication, that includes one of the coauthors, performed protein profiling of CSF samples from 13 LNB patients using Olink panels. Were samples from these patients used in both studies? If yes, please provide the information and how do they compare between platforms.

Also, I find the mentions of possible differences because of geographic locations or age bodes poorly for an approach that is being proposed for a possible clinical use.

Regarding lines 408-410, about a less invasive approach to be particularly beneficial for children, a 2024 publication, that includes one of the co-authors of this manuscript, supports my early comment that current serological tests already have high predictive value in patients with clinical signs of LNB. PMID: 39221969

Methods:

Please provide the method used for the *B. burgdorferi*-specific intrathecal antibody test.

Rebuttal:

Regarding the author's responses to two of my previous questions:

R2: This is shown by the results of the diagnostic classifier in plasma, where the model was only able to differentiate LNB patients from controls (and that with AUC of 0.80 and MCC of 0.48), performing much worse than what is expected for serologic assays.

Author response: We respectfully disagree and believe that an AUC of 0.8 for diagnosing LNB on a blood test is clinical relevant and is at least comparable to the diagnostic accuracy of serologic assays as reported in the ECDC systematic review from 2016 (Leeflang et al. BMC Infect Dis. 2016 doi: 10.1186/s12879-016-1468-4).

R3: The AUC does not inform about positive predictive value and negative predictive value. The MCC is more informative in terms of predictive value, and it was much lower, at 0.48.

AND

R2: To say that these results support a "diagnostic blood-based assays for LNB" and "have clinical relevance" is an overstatement. It is unclear to me how the investigators can claim improvement in diagnostic using this more expensive and resource intensive approach, or that this could be a valuable clinical tool when there is no evidence that this offers any benefit comparing with current methods. Their answer is that this would be addressed in a prospective study, but this is not true. This analysis can easily be done using current antibody assays on the samples tested by proteomics, as well as using laboratory data that already exists for all the patients and controls, as I am sure all have had Lyme disease serology performed as part of the evaluation. This comparison should be added to this study.

Author response: We acknowledge the overstatement and have thoroughly revised the entire text. What we wanted to express was that we believe our exploratory results are promising enough to foster further investigations. This has now been rephrased in the manuscript. Due to the retrospective design of this study we do unfortunately not have access to the serology results. We have clearly pointed this out in the revised text and underlined that future investigations should include this comparison.

R3: While this is a retrospective study, all samples have values for the *B. burgdorferi*-specific intrathecal antibody test. These results include both the serum and CSF testing. Therefore, the authors can use these results for the comparison with the models and add this information to the manuscript.

(Remarks on code availability)

REVIEWER COMMENTS

Reviewer #1 (Remarks to the Author):

Comments to the authors

The manuscript presented by Bach Nielsen and co-authors investigates machine learning-assisted MS-based proteomics for a rapid and accurate diagnosis of Lyme neuroborreliosis. Any tool, any new method that will improve the diagnosis of this nervous system infection is very valuable in the field of Lyme disease. In particular, the possibility of making a diagnosis from blood, in a less invasive way than a lumbar puncture, is very promising. One of the strengths of this work is that it was based on a large number of samples (483 CSF and 175 plasma) with development and validation cohorts. The analytical workflow is robust and the analytical equipments and softwares correspond to the state of the art for non-targeted proteomics. The manuscript is well written, the patients and methods thoroughly described and I have only some minor remarks.

Lines 142-143 and Figure 1: In the comparison control vs LNB, 41 proteins are significantly expressed with a fold change larger than 2. Quote Figure A (Volcano plot) and not Figure B (Overlap).

Author response: Thank you very much, this has been corrected in the text.

While the 10 proteins from the VM vs LNB comparison are visible on Figure 1B, 43 proteins (33+7+3) are recorded on Figure 1B for control vs LNB. What makes the difference between 41 and 43?

Author response: We thank the reviewer for spotting this mistake, the correct numbers are (31+7+3) = 41. This has been corrected in the figure accordingly.

Line 145: This sentence refers to Figure 1B, not Figure 1C

Author response: Thank you very much, this has been corrected in the text.

Line 202: The number of proteins identified in the plasma samples (median of 268 proteins) is quite low, which is not surprising in the absence of depletion or prefractionation, but only allows the more abundant proteins to be detected. The authors chose to use a very short 8 cm column (probably to increase the number of samples per day). What impact did this length of column have on the number of proteins identified, compared with previous plasma studies in the literature using much longer columns?

Author response: We appreciate the reviewer's observation. The use of an 8 cm column was, as the reviewer correctly guessed, a deliberate choice to prioritize throughput and robustness for future clinical implementation. However, we acknowledge that this shorter column, with its reduced chromatographic resolution, limits proteome depth compared to longer columns (25–50 cm) commonly used in plasma studies. Combined with the absence of depletion or fractionation, this contributed to the lower number of protein identifications, focusing detection on more abundant proteins. We have clarified this trade-off and its implications in the revised manuscript. Maybe the reviewer will find it interesting that we

compared our existing setup (sample prep and LC conditions including 8cm column) with the same setup where we measured on an Orbital Astral machine in stead of an Exploris, which increase the depth in plasma to around 800-1000 proteins.

Figure 3A: To be consistent note “Control vs LNB” on the Volcano plot and not “LNB control vs LNB”.

Author response: Thank you very much, this has been corrected in the figure.

Figure 3B: Line 205 states that 61 proteins are significantly different in VM vs LNB. However, on Figure 3B, a total of 68 proteins (28+33+4+3) are in the corresponding circles. Same remark for the 63 proteins stated Line 206 (LNB vs control), while a total of 69 proteins (33+30+3+3) are in the corresponding circles. What make these differences?

Author response: We apologize for this inconsistency, the text has been updated with the numbers matching the figure, the supplementary material and the code.

Line 194 (Fig 3B legend): As seen on Figure 3A and Supplemental Table 2, only 2 proteins (VWF, S100A9) exhibit a fold change larger than 2 in LNB vs control. Similar remark for VM vs LNB with only 6 proteins with a fold change larger than 2. There must be an error in the, since all the significant proteins are represented, and not only the significant proteins with a fold change larger than 2.

Author response: You are absolutely right, 3B contains all significant proteins and was not limited to proteins with large fold changes like 1B. The legend for Figure3B has been corrected.

Line 207: This sentence refers to Figure 3B, not Figure 3C

Author response: Thank you very much, this has been corrected in the text.

Given that, for plasma, all the significant proteins have been taken into account (and not just those that vary with a fold change larger than 2 as in CSF), are there any consequences for the robustness of a blood-based clinical test?

Author response: This is a very relevant question, thank you for your reflections! Proteins with large fold-changes and low variance would generally be preferable in a robust clinical assay. We acknowledge that the plasma-based model includes all statistically significant proteins, regardless of fold change, unlike the CSF model, which applies fold change threshold. We believe this difference reflects the distinct biological profiles of the two sample types. To mitigate potential overfitting and ensure robustness, we applied rigorous cross-validation, controlled for multiple testing, and evaluated model generalizability on an independent cohort. These steps help reduce the risk of including noise-prone biomarkers and

strengthen the clinical potential of the blood-based test. But we are also very aware that our findings need further validation before routine use.

Line 636 (Mat & Meth): As the MS equipment and the column are different from those used in the cited publications^{44,45,46}, the authors should give more details on LC conditions (e.g. gradient duration) and MS acquisition parameters (or cite additional publications).

Author response: We thank the reviewer for pointing this out. We have included more details on the sample preparation and LC MS setup in the methods section.

Reviewer #2 (Remarks to the Author):

The study explores the potential of mass spectrometry-based proteomics as a novel diagnostic tool for Lyme neuroborreliosis (LNB) in CSF and plasma of adults. The authors' stated aim is to develop a test for early diagnosis of LNB, avoiding the need for a lumbar puncture.

They analyzed stored CSF and plasma samples from adult patients with LNB, viral meningitis, controls, and other Lyme borreliosis manifestations and used machine learning analyses to identify panels of proteins that could distinguish between groups.

There are issues with this work. The first is the definition of the samples, in particular LNB and controls. LNB was defined based on a first-time positive *B. burgdorferi*-specific intrathecal antibody test and the ICD-10 code A69 for Borreliosis (which includes all presentations). There is no other information on the clinical presentation or information on pleocytosis.

Author response: We agree that additional information on clinical presentation and pleocytosis from the included LNB patients would have enabled a more accurate categorisation of LNB patients in accordance with the EFNS diagnostic criteria for 'probable' and 'definite' LNB. Because the study was performed on biobank samples, we did not have sufficient data to perform this subclassification. However, as mentioned in the discussion (page 17, lines 376-377), we believe a first-time positive *Bb* IT antibody index is well-validated as a simple, yet stringent and robust identifier of acute LNB patients. CSF leukocyte counts were available from 39/155 patients with LNB (mean of 155×10^6 cells/L) and from 22/127 patients with VM (mean of 172×10^6 cells/L). We initially chose not to include these in the manuscript due to the large number of missing values. This has now been added to the case definitions section (page 28, lines 662-663 and page 28, line 673).

The controls were individuals who had no pleocytosis and a negative *B. burgdorferi* specific intrathecal antibody index after being investigated for suspected LNB. There is no information if these patients had another diagnosis (of Lyme borreliosis or other disease). It is unclear if these controls would not include patients with other presentations of Lyme borreliosis.

Author response: Thank you for this comment. As this study was retrospective and based on available biobank samples we unfortunately did not have access to supplementary information on all participants.

However, we have now reviewed available patient records and extracted final diagnoses on 122/169 of the included control samples. The final diagnoses in this group were; Lyme borreliosis (erythema migrans, and post-treatment-Lyme-disease-syndrome) (n=26), other infectious diseases (EBV, COVID19, influenza, sinusitis, tonsillitis, rickettsiosis, and presumed viral infection without specified etiology) (n=28), non-infectious inflammatory conditions (rheumatoid arthritis, temporal arthritis, monoclonal gammopathy of undetermined significance (MGUS) and b-cell lymphoma) (n=4), neurological conditions (Bell's palsy, Parkinson's disease, migraine, tension headache and peripheral neuropathy) (n=16), degenerative condition of the spine (spinal stenosis, spinal disc herniation) (n=5). In the remaining 43 cases, no final diagnostic conclusion was reached. This additional information has now been added to the case definitions section (page 28-29, lines 676-685), as well as the additional approval for patient record review from the ethical committee (page 27, line 632).

The requirement of a positive IT antibody test anchors the LNB group to this definition. This is reflected on the list of differentially expressed proteins with an absolute log₂ fold change larger than 1 provided in in supplementary table 1. I rerun the list using online available tools, and found that 5 of 7 upregulated proteins in LNB compared with viral meningitis (VM) are within the GO CC term immunoglobulin complex GO:0019814, one to IgA immunoglobulin complex, and the other is Fc gamma binding protein. Similar findings are seen compared with controls, where most proteins are related to immunoglobulin complex and related terms. Also shown on the "signature of LNB" where "immunoglobulins were generally upregulated, while proteins involved in innate immune responses and neuroendocrine signaling were relatively downregulated in LNB compared to viral meningitis and controls". These issues are not discussed in the manuscript.

Author response: We thank the reviewer for noticing this. Proteins associated with humoral immune responses indeed clearly dominate the group of most significantly different abundances between LNB and VM/controls. The dominance of antibody-related proteins in the diagnostic models, could reflect that infections with different microorganisms cause pathogen-specific humoral immune responses that, to some degree, can be decoded using proteomics and ML assisted classification. We have now added this additional reflection in the discussion section (page 15 lines 306-309). We chose not to elaborate further on the potential pathophysiological role of different neuroendocrine signalling molecules as we felt such speculations were outside the aim of this study. We do however agree that exploration of these largely unknown pathophysiological processes in LNB would be extremely interesting.

The requirement of positive index excludes patients with possible LNB by EFNS criteria who present with pleocytosis and typical clinical presentation. Studies have shown that these patients have a shorter duration of symptoms compared with patients with positive IT index, therefore representing an earlier phase of illness.

Author response: Thank you for bringing this up. We fully agree with this important point that we hope to be able to address in a prospective validation study that we are actively pursuing. We have now added this relevant point to the discussion section (page 17-18, lines 377-379).

The fact that CXCL13 was not found to be increased in CSF of LNB patients, or to be part of the models is highly problematic. While the explanation that “the limited detection range inherent to this approach means that certain low-abundance but highly disease-specific proteins, such as CXCL13, may not be captured. Given that previous proteomic studies in LNB have also failed to detect CXCL13, this highlights a general challenge in proteomic biomarker discovery rather than a limitation of our study” is possible, it does not recommend the methodology for discovery. CXCL13 is measured in higher levels in CSF in acute LNB than in other neuroinflammatory diseases and may precede antibody production. It has been proposed as both an early diagnostic marker and a marker of treatment response.

Author response: CXCL13 is a valuable and well-validated biomarker in LNB. However, as also quoted above, because we chose to limit the spectrometry-range we would not expect to identify CXCL13 or other chemokines due to their low relative abundances.

It is important to note that this study was designed to assess the diagnostic potential of a disease specific ‘fingerprint’ of protein expressions. In this study, we deliberately prioritised a short turnaround time of the analysis over extended range that could potentially have allowed us to identify very low abundant cytokines and chemokines like CXCL13. We did this because we wanted to test if MS based proteomics could be a feasible and realistic alternative to the current diagnostic modalities in LNB. In a clinical context a short turnaround time would be highly valuable both to the clinician and to the patient. Extending the range would significantly increase the turnaround time and make the test unfeasible to use in a routine clinical setting. In fact, we believe it should rather be considered a strength of the results that the relatively high level of diagnostic accuracy is achieved even without the inclusion of CXCL13 in the model. We have rephrased the paragraph on this matter in the manuscript (page 18 line 387-392).

In terms of the ML, twelve models were tested for the CSF data and for plasma data. A LR model was chosen for LNB vs viral meningitis, and an SVC model for LNB vs. controls. No models were developed using the 3 groups. The LR model for LNB vs VM had a MCC of 0.81 on the test set and 0.7 on the validation. For LNB vs controls, the SVC model had an MCC of 0.76 on the test set and 0.63 on the validation. While these are good levels of accuracy, I would disagree with the statement that “Overall, MS-based proteomics coupled with ML demonstrated strong potential as a diagnostic tool for LNB, with particular proteins playing key roles in distinguishing between conditions.” As the LNB group was defined by positive IT index, this would have a MCC of 1. It is also highly likely that Lyme serological test would also have a higher MCC, and possibly CXCL13.

Author response: Thank you for your comment. The aim of our study was to develop blood-based diagnostic assays for LNB, meaning that patients suspected of having LNB would no longer require a lumbar puncture to confirm the diagnosis. Today there is no diagnostic blood-based assays for LNB, and we believe the findings presented are of clinical relevance.

We agree that the performance of proteomics as a diagnostic test for LNB in plasma needs to be compared head-to-head with existing diagnostic alternatives. This will be part of a prospective validation study that we are planning.

The plasma analysis showed that VM patients and controls had higher immunoglobulin levels than LNB patients. These are highly unexpected and unexplained results, particularly comparing with controls. An explanation for these findings, including technical issues need to be provided.

Author response: Thank you for drawing attention to these interesting findings. There are many interesting patterns of protein abundances that inspire to further investigation, analyses and interpretations throughout this fairly large dataset. We agree that we could have elaborated more upon this specific finding in the discussion. It is important to note that the reported relative abundances of proteins are not equal to protein concentrations. The observed higher relative abundances of immunoglobulins in plasma from patients with VM and controls compared with LNB are likely the result of pathophysiological differences between the groups. Thus, patients with LNB (especially those with a positive *Bb* IT index as included in this study) show a clear compartmentalisation of the humoral immune response to the CNS, patients with VM may be more likely to present in the acute phase where the immune response is more systemically distributed, and the group of non-LNB controls included patients with 'peripheral' presentations of borreliosis, rheumatological and other systemic conditions as the cause of the symptoms that initially gave rise to the suspicion of LNB.

We have expanded our discussion on this particular matter (page 15, lines 306-309, and page 16 lines 327-336).

The ML model for plasma focused only in differentiating between controls and LNB. The SVC model chosen had a MCC of 0.83 on the test set, but only 0.48 in the validation set.

The authors stated that the model identified acrodermatitis chronica atrophicans and erythema migrans compared to post-treatment Lyme disease syndrome and viral meningitis. However, little detailed information is given regarding the individual presentations, the numbers are small for some, and no comparison is done to available methodologies.

Author response: The definitions of clinical Lyme borreliosis manifestations were in line with published European guidelines and case definitions.

The following case-definitions were used:

Erythema migrans

We defined erythema migrans as early localized Lyme borreliosis, solely from the clinical presentation

Post-treatment Lyme disease syndrome

If symptoms, related to a previously verified and treated Lyme borreliosis manifestation, persisted for more than 6 months, the condition was termed post-treatment Lyme disease syndrome.

Acrodermatitis chronica atrophicans

Acrodermatitis chronica atrophicans was diagnosed, if the clinical presentation was compatible with acrodermatitis chronica atrophicans and elevated serum *B. burgdorferi* s.l. IgG antibodies were detected. If possible, a skin sample was obtained for histological examination to further confirm the diagnosis.

All patients in this cohort were evaluated at the Unit for Tick Borne Infections which is an outpatient clinic under the Department of Infectious Diseases at Copenhagen University Hospital, Rigshospitalet. Patients were assessed by two experienced infectious diseases specialists.

We have now clarified and expanded the case definitions in the 'materials and methods' section (page 29, lines 686-703).

Another difficulty is the possible variation caused by age differences in the groups. While there is no analysis of the differences, by the information provided (median age and IQR) it is possible that there are significant differences between the groups. This is mentioned only in brief in the discussion.

Author response: We agree that there is a possibility that the age difference may have some effect on the immunological patterns that we detect, but as mentioned in the limitations section we deliberately chose to prioritise control populations that resembled real-life as closely as possible over a sex and age-matched case control design. We believe it is fair to say that there are relevant pros and cons to both approaches. We have now elaborated further on this issue in the discussion (page 18, lines 399-402).

The discussion highly overrepresents the results, and downplays the problems and limitations of the study. Examples include statements like "MS-based proteomics can accurately differentiate patients with LNB from patients with viral meningitis and non-LNB controls", "observed a high level of diagnostic accuracy in plasma, highlighting the potential of blood-based proteomics as a clinically valuable tool" and "In conclusion, our findings demonstrate that MS-based proteomics can accurately distinguish LNB from controls in both CSF and plasma". The results do not support these statements. It is likely that the models would not perform any better than current antibody-based assays.

Author response: Thank you for this important critique and in response to your comment, we have revised the discussion to use more measured and precise language that better reflects the scope and limitations of the current study. It is important notice that there is a current lack of diagnostic blood-based assays for LNB and here we believe the findings presented are of clinical relevance.

Figures are difficult to read due to small print, particularly figures 1 and 3. There are label overlaps in Figure 1 and supplementary figures 2 and 3.

Author response: Thank you for this valuable feedback. We fully agree that the readability of Figures 1 and 3 is challenged by the small font size. Unfortunately, due to the high number of significant proteins

shown, it is not feasible to increase the font size without severely compromising the clarity, structure or size of the figure. We've aimed to strike a balance by including as much informative detail as possible, but we recognize the limitations and are open to guidance on alternative formatting options if available.

Regarding label overlap in Figure 1 and Supplementary Figures 2 and 3, this issue stems from the density of significant features. We acknowledge this limitation and have move names manually, but for Suppelementary Figure 2, we had to move the threshold for including labels to log2fc of 2 and -2, to avoid overlapping labels.

Reviewer #3 (Remarks to the Author):

The manuscript by Nielsen et al reports the use of an MS-based proteomics technique for Lyme neuroborreliosis in adults. This is quite an interesting paper expanding on the value of host-response profiles for predicting disease. I have some comments/questions for consideration:

(1) The authors should be praised for having a diverse sample set including non-Lyme diseases. For the viral meningitis cases, what assay was used for pathogen detection. Should state the assay. For example if BioFire FilmArray ME Panel was used, the HSV target has some limitations. In clinical practice, HSV testing may need to be tested on a separate assay, due to concerns on assay performance.

Author response: Thank you for mentioning this. We use BioFire as our standard multiplex PCR assay. It is generally applied on CSF from all patients with a pleocytosis and a suspected CNS infection. Due to the suboptimal sensitivity for HSV, a specific HSV PCR is conducted if BioFire is negative and HSV is clinically suspected. This has now been specified in the manuscript (page 28, lines 667-670).

(2) On the same topic, the authors noted that a lot of the signal detected MS in CSF were immunoglobulins. Can the authors suggest, perhaps in the Discussion Section, how other diseases such as multiple sclerosis which has unique immunoglobulin profiles in CSF, would impact the assay performance?

Author response: Thank you for this question. We have added a note in the discussion section addressing this point (page 18, lines 380-383). While multiple sclerosis (MS) also presents with distinct immunoglobulin profiles in CSF, we believe that the broader proteomic signature—including non-immunoglobulin proteins—would still allow for discrimination between LNB and MS. That said, even if discrimination between LNB and MS were limited, this may have minimal impact on clinical utility, as the two conditions typically present with distinct clinical symptoms and are rarely confused diagnostically. Nonetheless, it would be interesting in future work to evaluate how well the model performs in more nuanced cases, such as patients with co-existing MS and suspected CNS infections.

(3) Can the authors comment on how long the test takes? This helps discuss how translatable it is for real world practice if this is ever adopted for routine clinical testing. I see a comment of 2 hours in line 90. Is

this true, in the context of total testing time, not just the mass spec phase and/or informatics. What is the turnaround time from sample to fully verified result?

Author response: Thank you for highlighting this important point regarding clinical translatability. The previously mentioned “2 hours” was intended to illustrate the potential of streamlined proteomic workflows. Our sample preparation protocol is set up for 96-well plates, where the total sample preparation takes 5hr and 25 minutes + 21minutes LC/MS gradient + 21 minutes data analysis → approximately 6hours, but further automation and optimization could make substantially shorter processing times feasible. We have edited the text and provided a more accurate description of our present turnaround time in the methods section.

Even without further optimizations, we believe this level of throughput already represents a significant improvement over current clinical methods, which often require days to weeks, and demonstrates promising potential for future clinical implementation.

Importantly, this workflow is already implemented within the Department of Clinical Biochemistry at the hospital, which processes over 30,000 diagnostic tests daily. This is not a research setting but a fully integrated clinical laboratory, meaning the infrastructure, personnel, and quality assurance systems needed for routine implementation are already in place. Therefore, while further optimization is certainly possible, we believe the path to real-world application is relatively short, given the existing integration into routine clinical workflows.

(4) For machine learning (ML), how did the authors select the 12 models? At least logistic regression (LR) and support vector machine (SVM) was used. Was the use of LR and SVM based on a priori experience, familiarity, or other? The reason is there's been some studies showing an agnostic approach using a range of ML techniques could identify better models. Automated ML software has this advantage, although, it won't replace data scientists, some studies using MALDI-TOF-MS for infectious disease host responses found automated ML systems finding better models compared.

Author response: Thank you for this valuable comment. The 12 models were selected from the scikit-learn library in Python to represent a diverse set of machine learning approaches, including linear models, support vector machines, tree-based methods, nearest neighbors, neural networks, and ensemble techniques. Our intention was to take a data-driven approach to model selection, as the reviewer also suggests. Rather than relying solely on prior familiarity, we systematically evaluated model performance across this range and selected the best-performing model based on cross-validation metrics. This strategy aligns with principles used in automated ML frameworks, while still allowing us to retain interpretability and control over preprocessing and evaluation steps. We have expanded the description in the methods section. Please also see the model comparisons in supplementary figure 4 and 5.

(5) Often, ML models should be further validated with a tertiary dataset, which helps with generalization. Can the authors comment on how the results from this initial study are likely to be translated to samples from another location? For instance, disease prevalence is a limitation here as the

authors noted for viral meningitis. If a new dataset were to have a higher prevalence of different diseases, could this alter the results?

Author response: We are very excited to see how these findings translate to external cohorts and really appreciate the reviewer's important point regarding the need for validation. In this initial study, we validated our models on an independent Danish cohort, which provides a solid first step toward establishing robustness. However, we fully acknowledge that geographic, ethnic, and disease prevalence differences may affect model performance. We are actively pursuing further validation efforts in more diverse populations to assess generalizability across different settings. While it is difficult to predict the exact impact of varying disease prevalence or population characteristics at this stage, we agree that such external validations will be essential before clinical translation and are a key part of our future work. We have expanded the discussion on generalizability and hope it reflects these important considerations better.

(6) For ML models, are there any other variables used to evaluate the model such as F-statistic and/or Brier Scores

Author response: We primarily reported the Matthews correlation coefficient (MCC), as it summarizes overall classification performance by incorporating all elements of the confusion matrix. In addition, we used the weighted F1 score during cross-validation for hyperparameter tuning, ensuring that precision and recall were balanced across classes. We did not directly calculate the Brier scores, but we did test the probability calibration with a logistic regression (integrated in scikit-learn's CalibratedClassifierCV) and found that additional calibration steps did not improve performance, suggesting that the predicted probabilities were already reasonably well-calibrated. We thank the reviewer for the detailed insight and have elaborated on our methods section.

REVIEWER COMMENTS

Reviewer #2 (Remarks to the Author):

REVIEWER COMMENTS

This is the 2nd review of this study, which explores the potential of mass spectrometry-based proteomics as a novel diagnostic tool for Lyme neuroborreliosis (LNB) in CSF and plasma of adults. Again, the stated aim is to develop a test for early diagnosis of LNB, avoiding the need for a lumbar puncture.

The issues raised on my previous review are still unresolved. The main one is that as written, the data does not support the claims. This starts with the title, which claims a rapid and accurate diagnosis of Lyme neuroborreliosis in adults using mass-spectrometry-based proteomics and continues in the abstract and the discussion. The major issue is that this approach is unlikely to be any better than current conventional serology.

Author response: We have substantially revised the title, abstract, aim and discussion to reflect a more balanced and accurate aim and interpretation of the results. The title no longer claims a rapid and accurate diagnosis, the abstract and aim has been adjusted to reflect that the intention was to characterize the molecular profiles and to explore if machine learning-assisted proteomics holds a potential in the development of new diagnostic tools in LNB. In the discussion we have tempered the conclusions and made it clear in the perspectives that

a diagnostic tool based on a broader panel of host-response proteins could potentially alleviate some of the diagnostic challenges in LNB.

This is shown by the results of the diagnostic classifier in plasma, where the model was only able to differentiate LNB patients from controls (and that with AUC of 0.80 and MCC of 0.48), performing much worse than what is expected for serologic assays.

Author response: We respectfully disagree and believe that an AUC of 0.8 for diagnosing LNB on a blood test is clinical relevant and is at least comparable to the diagnostic accuracy of serologic assays as reported in the ECDC systematic review from 2016 (Leeflang et al. BMC Infect Dis. 2016 doi: 10.1186/s12879-016-1468-4).

The models were no able to differentiate LNB from other LB patients (EM, ACA). This is not different from current antibody-based tests.

Author response: We did not test the model's discriminatory power between LNB and other LB patients, but the predicted probabilities for patients with active LB infection was significantly higher than patients with inactive infection. We believe that future development could benefit from this finding and that broader protein profiles might be relevant to consider in such developments.

To say that these results support a "diagnostic blood-based assays for LNB" and "have clinical relevance" is an overstatement. It is unclear to me how the investigators can claim improvement in diagnostic using this more expensive and resource intensive approach, or that this could be a valuable clinical tool when there is no evidence that this offers any benefit comparing with current methods. Their answer is that this would be addressed in a prospective study, but this is not true. This analysis can easily be done using current antibody assays on the samples tested by proteomics, as well as using laboratory data that already exists for all the patients and controls, as I am sure all have had Lyme disease serology performed as part of the evaluation. This comparison should be added to this study.

Author response: We acknowledge the overstatement and have thoroughly revised the entire text. What we wanted to express was that we believe our exploratory results are promising enough to foster further investigations. This has now been rephrased in the manuscript. Due to the retrospective design of this study we do unfortunately not have access to the serology results. We have clearly pointed this out in the revised text and underlined that future investigations should include this comparison.

Most of the discussion should be rewritten as it continues to overrepresents the results and downplays the problems and limitations of the study. Again, the current data does not support that this approach is a clinically valuable tool, or that it could be used for initial diagnosis and monitoring of treatment response, or that could distinguish Lyme neuroborreliosis any better than current assays (which are much cheaper and faster). While there are many parts of the discussion that need to be modified, in particular, lines 284-289, 370-384, 397-400

It would be best if the authors focused on the exploratory aspects of the research.

Author response: We thank the reviewer for this suggestion and have now heavily rewritten the discussion focusing on the exploratory aspects of our research. We have tempered the data-interpretation and the clinical implications as well as expanded the discussion on limitations trying to make it clear that this study and the results it produced should be considered as exploratory and that they need further investigations and external validation before determining any direct clinical implications.

An issue that has not been answered is that the findings of many increased protein related to immunoglobulins in Lyme neuroborreliosis CSF are likely to due to the requirement of a positive IT antibody test for LNB. The answer provided and the text added to the manuscript do not address this core issue and reflect a lack of understanding of the definition bias. The only way to ascertain the effect of the positive IT an index requirement in the definition would be to compare the findings on probable LNB patients who had clinical features and pleocytosis but not positive IT ab index. This potential bias should be discussed.

Author response: We agree with this important observation and have added it to the discussion and pointed out that it is unknown whether a similar proteomic signature is detectable in patients with probable LNB, with compatible clinical features and pleocytosis but lacking a positive antibody index. Even if this is not the case, it does not mean that we would necessarily expect machine learning-assisted proteomics to achieve a less accurate diagnostic performance in the population of early LNB patients. If the model was trained on proteome data from these patients, the resulting best classifier could differ from the one we found including a somewhat different protein pattern – potentially with a less pronounced dominance of immunoglobulin-related proteins.

While the authors added that they were “likely to have missed some patients with early LNB” (line 394), a more detailed discussion of this issue should be added. While the positive Bb IT antibody index is likely a validated identifier of LNB in the correct setting, it is a later finding, as it has been shown to correlate with duration of illness. As the stated aim is for an early diagnostic test, the lack of this group (clinical picture + pleocytosis but negative IT) and the implications for the study should be more clearly stated.

Author response: We fully agree with the reviewer. It would have been ideal to have the full range of disease duration of LNB patients included, but due to the retrospective design of this exploratory study, this was unfortunately not possible. In addition to revising the aim, we have expanded the discussion on this limitation and encouraged future developments in this direction.

As the study does not have patients with meningitis, encephalitis, multiple sclerosis and only one control with CNS lymphoma, there is no data to support the added text (lines 397-400) “Thus, the demonstrated ability of proteomics to differentiate LNB from VM is important for the applicability of the test in settings where other inflammatory CNS conditions are differential diagnoses (meningitis, encephalitis, CNS lymphoma, multiple sclerosis etc.)”

Author response: We apologise for not stating clearly enough that this sentence refers to the results from CSF, where a large cohort of patients with viral meningitis of different aetiologies is included, and not plasma.

I also take issue with some of the author's response to reviewer 3.

2-While typical acute LNB and multiple sclerosis have mostly distinct clinical features, the aim would be for a general test and false positive results would have major impact on clinical utility.

Author response: We agree with this point.

3-I am not sure what is the basis for the author's statement that "even without further optimizations, we believe this level of throughput already represents a significant improvement over current clinical methods, which often require days to weeks, and demonstrates promising potential for future clinical implementation." Current assays for antibody-based diagnosis of Lyme borreliosis DO NOT require days to weeks as stated by the authors. These are EIA-based tests that can be rapidly performed once the sample was received. Even immunoblots can be performed much faster than proteomics. Are the authors adding the transport time to a central laboratory in their assumption and/or time due to test batching?

Author response: Turnaround times vary from hospital to hospital depending on the availability of diagnostic services, the local set-up for assay capacity, need for batching, transport of samples etc. We have revised the text so that it neither insinuates unrealistically optimistic turnaround times for proteomics nor reports pessimistic turnaround times for existing assays.

Reviewer #3 (Remarks to the Author):

Thank you for addressing the questions/comments. I feel the authors have adequately addressed them and I have nothing further to add.

REVIEWERS' COMMENTS

Reviewer #2 (Remarks to the Author):

This is the 3rd review of this study, which explores the potential of mass spectrometry-based proteomics as a novel diagnostic tool for Lyme neuroborreliosis (LNB) in CSF and plasma of adults. The manuscript is improved, but there are still points to be addressed.

Author response: We thank the reviewer for the time and effort the reviewer has put into reviewing our manuscript again. We are thankful for the input and believe that the comments have helped improve the manuscript.

My comments:

Abstract: while Lyme neuroborreliosis can be severe, most cases are not (when compared with bacterial or fungal meningitis). I would remove the adjective.

Few patients with LNB will require "continuous monitoring of disease activity". I would remove this from the abstract

Author response: We have removed "severe" from the abstract.

Introduction:

Lines 74-75. Measurement of intrathecal antibody production for diagnosis of LNB is a requirement only in Europe. This is not required in the US. This need to be clarified in the text. There should also be a discussion of the differences in CSF testing between US and Europe, as a positive intrathecal antibody index is thought to be less common in the US.

Author response: We thank the reviewer for this important clarification. Our original intent in lines 74-75 was to emphasize the invasive nature of lumbar puncture rather than to comment on regional diagnostic criteria or the use of the intrathecal antibody index. We have revised the text accordingly (now lines 68-69).

Discussion:

Line 282-286. I am confused about the comments on the study by Angel et al. What do the authors mean by:

"However, the focus on other Lyme disease manifestations rather than LNB likely reflects the, somewhat debated, geographical difference in prevalence of neurological involvement with a *B. burgdorferi* s.l. strain-dependent lower incidence in North America compared to

Europe 18,19. The sentence as written does not make sense. It appears to be truncated. The study only included LNB patients (as all had CSF pleocytosis but 1 patient that had signs of CNS involvement), so the comment regarding other manifestations is unclear.

Author response: We thank the reviewer for pointing out this confusion and apologize for the truncated sentence, which must have occurred during revision. The geographical context is not relevant and the comment regarding other manifestations is unclear - both have been removed. We appreciate that the reviewer has drawn our attention to this part of the discussion, including points further down in this document, and have therefore carefully revised the section (lines 220-227) and expanded our results section with a comparison of the findings from the different studies (lines 192-208).

The citations in the text are confusing. I would prefer that the reference number comes soon after the citation, and not at the end of the sentences.

Author response: We thank the reviewer for this helpful suggestion. The reference numbers have been moved so that they appear immediately after the respective citations, as recommended.

I am also confused by the information in parenthesis. Is it necessary to say Angel et al. (2013), Fredriksson et al. (Sweden, 2024), Gęgotek et al. (Poland, 2024)?

Author response: We thank the reviewer for highlighting this. The country and year are irrelevant and have been removed.

I find the discussions of these previous studies underwhelming. I would expect at least a discussion of the comparison of the data from this study to the other studies in LNB, particularly references 17, 20 and the cohort with acute LNB from reference 22. These comparisons can be provided in detailed as supplemental information.

Author response: We thank the reviewer for this very helpful suggestion. We have now compared our results with the studies cited (Angel, Fredriksson, Gęgotek, and Nilsson). The complete protein lists from each study are provided in Supplementary Table 4, and the comparisons are presented in Figure 5C, Supplementary Table 4 and described in the extended results section (lines 192-208).

Also, a recent publication, that includes one of the coauthors, performed protein profiling of CSF samples from 13 LNB patients using Olink panels. Were samples from these patients used in both studies? If yes, please provide the information and how do they compare between platforms.

Author response: No, the patients were not part of the present study.

Also, I find the mentions of possible differences because of geographic locations or age bodes poorly for an approach that is being proposed for a possible clinical use.

Author response: We appreciate the reviewer's observation. We mentioned age and geographical factors to highlight some of the limitations of our study and to acknowledge potential sources of variability. These factors were not investigated in the present exploratory analysis, so it remains to be determined whether they influence the performance of our model. We agree that this will be important to address in future work before any potential clinical implementation.

Regarding lines 408-410, about a less invasive approach to be particularly beneficial for children, a 2024 publication, that includes one of the co-authors of this manuscript, supports my early comment that current serological tests already have high predictive value in patients with clinical signs of LNB. PMID: 39221969

Author response: We thank the reviewer for highlighting the recent study by Bloch et al. As noted, this work demonstrated a high predictive value of borrelia serology in children with facial nerve palsy, a clinically important subgroup. At the same time, we believe they do not preclude a potential role for proteomics in the broader diagnostic work-up.

Methods:

Please provide the method used for the *B. burgdorferi*-specific intrathecal antibody test.

Author response: The *Borrelia* specific intrathecal antibody index was measured using a second-generation, flagella antigen-based capture enzyme immunoassay (IDEIA Lyme Neuroborreliosis, Oxoid, Hampshire, UK). The test result was expressed as a specific capture optical density (OD) value in CSF/OD-serum \times (OD-CSF- OD-serum). An elevated (positive) index was assumed if the specific capture IgG and/or IgM index was > 0.3 in combination

with ODCSF ≥ 0.15 . This paragraph has now been added to the Methods section lines 391-396.

Rebuttal:

Regarding the author's responses to two of my previous questions:

R2: This is shown by the results of the diagnostic classifier in plasma, where the model was only able to differentiate LNB patients from controls (and that with AUC of 0.80 and MCC of 0.48), performing much worse than what is expected for serologic assays.

Author response: We respectfully disagree and believe that an AUC of 0.8 for diagnosing LNB on a blood test is clinical relevant and is at least comparable to the diagnostic accuracy of serologic assays as reported in the ECDC systematic review from 2016 (Leeflang et al. BMC Infect Dis. 2016 doi: 10.1186/s12879-016-1468-4).

R3: The AUC does not inform about positive predictive value and negative predictive value. The MCC is more informative in terms of predictive value, and it was much lower, at 0.48.

Authors response: We thank the reviewer for this comment and agree that MCC provides complementary information to AUC regarding predictive values. In our dataset, a MCC (range from -1 to 1) of 0.48 corresponds to a PPV of 0.53 and an NPV of 0.93. For comparison, using the data reported in the ECDC systematic review by Leeflang et al. we calculate a MCC of 0.41 for the serologic assay, with PPV and NPV values of 0.31 and 0.97, respectively. Thus, while our model shows a slightly lower NPV, it demonstrates substantially higher PPV and overall balance (as reflected in MCC) compared to serology. Importantly, PPV is a critical parameter in clinical diagnostics, as it directly reflects the likelihood that a positive test truly indicates disease. We therefore consider our results to be within a clinically relevant range and, in view of the markedly improved PPV, a potential improvement over existing serological approaches. We have added lines 298-301 to cover this important point.

AND

R2: To say that these results support a "diagnostic blood-based assays for LNB" and "have clinical relevance" is an overstatement. It is unclear to me how the investigators can claim improvement in diagnostic using this more expensive and resource intensive approach, or that this could be a valuable clinical tool when there is no evidence that this offers any benefit comparing with current methods. Their answer is that this would be addressed in a

prospective study, but this is not true. This analysis can easily be done using current antibody assays on the samples tested by proteomics, as well as using laboratory data that already exists for all the patients and controls, as I am sure all have had Lyme disease serology performed as part of the evaluation. This comparison should be added to this study.

Author response: We acknowledge the overstatement and have thoroughly revised the entire text. What we wanted to express was that we believe our exploratory results are promising enough to foster further investigations. This has now been rephrased in the manuscript. Due to the retrospective design of this study we do unfortunately not have access to the serology results. We have clearly pointed this out in the revised text and underlined that future investigations should include this comparison.

R3: While this is a retrospective study, all samples have values for the *B. burgdorferi*-specific intrathecal antibody test. These results include both the serum and CSF testing. Therefore, the authors can use these results for the comparison with the models and add this information to the manuscript.

Author response: We thank the reviewer for the opportunity to clarify. Although we would have liked to perform this comparison, we unfortunately do not have the requested values. In Denmark, intrathecal antibody indexes are measured using the capture ELISA assay (IDEIA, Lyme Neuroborreliosis, Oxoid, Hampshire, UK), which reports the ratio of *Borrelia*-specific IgG/IgM in serum and CSF, but not absolute antibody concentrations (Now described in further details in the Methods section 391-396).

While some LNB patients may have had separate serological testing performed, these data were not available to us due to the retrospective study design and data access restrictions, which covered only proteomic data and limited anonymized metadata. We have described this limitation in the manuscript and plan to address it in future studies.